# A non-linear system patterns Rab5 GTPase on the membrane

Alice Cezanne[1], Janelle Lauer[1], Anastasia Solomatina[1,2,3], Ivo F Sbalzarini[1,2,3], Marino Zerial[1]*

[1]Max-Planck Institute of Molecular Cell Biology and Genetics, Dresden, Germany; [2]Chair of Scientific Computing for Systems Biology, Faculty of Computer Science, Dresden, Germany; [3]MOSAIC Group, Center for Systems Biology Dresden, Dresden, Germany

**Abstract** Proteins can self-organize into spatial patterns via non-linear dynamic interactions on cellular membranes. Modelling and simulations have shown that small GTPases can generate patterns by coupling guanine nucleotide exchange factors (GEF) to effectors, generating a positive feedback of GTPase activation and membrane recruitment. Here, we reconstituted the patterning of the small GTPase Rab5 and its GEF/effector complex Rabex5/Rabaptin5 on supported lipid bilayers. We demonstrate a 'handover' of Rab5 from Rabex5 to Rabaptin5 upon nucleotide exchange. A minimal system consisting of Rab5, RabGDI and a complex of full length Rabex5/Rabaptin5 was necessary to pattern Rab5 into membrane domains. Rab5 patterning required a lipid membrane composition mimicking that of early endosomes, with PI(3)P enhancing membrane recruitment of Rab5 and acyl chain packing being critical for domain formation. The prevalence of GEF/effector coupling in nature suggests a possible universal system for small GTPase patterning involving both protein and lipid interactions.

## Introduction

Membrane compartmentalization is of central importance for a variety of biological functions at multiple scales, from sub-cellular structures to multi-cellular organisms. Processes such as cell polarization, protein and lipid sorting within sub-cellular organelles or cell and tissue morphogenesis depend on the emergence of patterns (*Turing, 1952*; *Halatek et al., 2018*). In *Caenorhabditis elegans*, symmetry breaking of the plasma membrane is caused by PAR proteins that sort into distinct anterior and posterior cortical domains and generate cell polarity (*Kemphues et al., 1988*; *Motegi and Seydoux, 2013*). In budding yeast, the site of bud formation is marked by a single, discrete domain of Cdc42 on the plasma membrane (PM) (*Ziman et al., 1993*; *Chen et al., 1997*; *Leberer et al., 1997*). In xylem cells, ROP11 is organized into multiple domains on the PM where it interacts with cortical microtubules to regulate cell wall architecture (*Yang and Lavagi, 2012*; *Oda and Fukuda, 2012*). Membrane compartmentalization is not limited to the plasma membrane but occurs also on cytoplasmic organelles. On early endosomes (EE), Rab5 exists in domains where it regulates vesicle tethering and fusion (*McBride et al., 1999*; *Sönnichsen et al., 2000*; *Franke et al., 2019*).

Cdc42, ROP11 and Rab5 are small GTPases, a class of molecules that play an important role in symmetry breaking and membrane compartmentalization. Small GTPases use GTP/GDP binding to act as an ON/OFF switch. The cycling between GTP and GDP-bound states is regulated by guanine nucleotide exchange factors (GEFs) and GTPase activating proteins (GAPs) (*Bos et al., 2007*; *Cherfils and Zeghouf, 2013*). Most small GTPases are post-translationally modified by lipid chains which allow them to associate with membranes (*Wang and Casey, 2016*). The inactive GTPase forms a high-affinity complex with guanine dissociation inhibitor (GDI), regulating membrane cycling

*For correspondence:
zerial@mpi-cbg.de

Competing interests: The authors declare that no competing interests exist.

(*Sasaki et al., 1990*; *Ghomashchi et al., 1995*; *Cherfils and Zeghouf, 2013*). Nucleotide exchange to GTP or a GTP analogue prevents interaction with GDI and frees the GTPase to interact with the membrane, where it can recruit effector proteins and mediate downstream activities (*Wu et al., 2010*; *Langemeyer et al., 2018*). Upon hydrolysis of GTP to GDP the GTPase is once again available for extraction from the membrane by GDI (*Rak et al., 2004*; *Ghomashchi et al., 1995*; *Pylypenko et al., 2006*).

It has been proposed that small GTPase patterning can arise from the coupling of GEF activity and effector binding (*Horiuchi et al., 1997*; *Zerial and McBride, 2001*). In this way, an active GTPase can recruit its own GEF, creating a local, positive feedback loop of GTPase activation and membrane recruitment. In general, self-organizing systems that form spatial patterns on membranes often exhibit such non-linear dynamics of membrane recruitment and activation (*Halatek et al., 2018*). The prevalence of GEF/effector coupling in small GTPase systems suggests that this may be a general mechanism for symmetry breaking and spatial organization of GTPases (*Goryachev and Leda, 2019*). The Rab5 GEF, Rabex5 is found in complex with the Rab5 effector Rabaptin5. (*Horiuchi et al., 1997*). Similarly, the Cdc42 GEF Cdc24 is coupled to the effector Bem1 (*Chenevert et al., 1992*). Computational modelling revealed a Turing-type mechanism of pattern formation by a minimal system composed of Cdc42, the Bem1/Cdc24 complex and GDI (*Goryachev and Pokhilko, 2008*; *Goryachev and Leda, 2017*). In plants, the ROP11 GEF, ROP-GEF4, forms a dimer that catalyzes nucleotide exchange but also interacts with the active ROP11 (*Nagashima et al., 2018*). We focus on Rab5, its GEF/effector complex Rabex5/Rabaptin5, and RabGDI (hereafter referred to as GDI) in order to investigate general mechanisms for the spatial organization of peripheral membrane proteins.

Rabex5/Rabaptin5 is one of the best characterized GEF/effector complexes in eukaryotes. Rabex5 is a 57 kDa Vps9 domain containing GEF for Rab5 (*Horiuchi et al., 1997*; *Delprato and Lambright, 2007*; *Lauer et al., 2019*). Rabaptin5 is a 99 kDa protein with multiple protein-protein interaction sites that colocalizes with Rab5 on EE and is essential for endosome fusion (*Stenmark et al., 1995*; *Horiuchi et al., 1997*). As Rabaptin5 forms a dimer in solution, the complex is a tetramer of two Rabaptin5 and two Rabex5 subunits (*Lauer et al., 2019*). The interaction with Rabaptin5 has been shown to increase Rabex5 GEF activity and produce structural rearrangements in Rabex5 (*Delprato et al., 2004*; *Delprato and Lambright, 2007*; *Lippé et al., 2001*; *Horiuchi et al., 1997*; *Zhang et al., 2014*; *Lauer et al., 2019*). By binding active Rab5, Rabaptin5 localizes the enhanced GEF activity of Rabex5 in the vicinity of active Rab5, thereby creating the positive feedback loop. In addition, Rabex5 can be recruited to EE via binding to Ubiquitin via two distinct Ubiquitin binding domains near the N-terminus (*Penengo et al., 2006*). Interestingly, Ubiquitin binding enhances GEF activity toward Rab5 helping to initiate the positive feedback loop on endosomes carrying ubiquitinated cargo (*Lauer et al., 2019*). *Blümer et al., 2013* observed that artificially targeting Rabex5 to mitochondria resulted in Rab5 recruitment to these organelles, suggesting that Rabex5 can be sufficient for localizing Rab5 to a membrane compartment. Rab5 associates with the membrane by two 20-carbon geranylgeranyl chains attached at the C-terminus of the protein (*Farnsworth et al., 1994*). Molecular dynamics simulations showed that both cholesterol and PI (3)P accumulate in the vicinity of Rab5, and predicted a direct interaction with PI(3)P mediated by an Arg located in the flexible hypervariable region (HVR) between the C-terminal lipidation and the conserved GTPase domain (*Edler et al., 2017a*).

Elucidating the precise mechanisms of self-organization of peripheral membrane proteins is critical to understanding endomembrane identity and functionality. We hypothesize that, similar to what has been observed for Cdc42 in silico, Rab5, Rabex5/Rabaptin5 and GDI comprise a minimal system that is capable of spatially organizing Rab5. We made use of in vitro reconstitution to test this hypothesis and elucidate the contributions of individual components to membrane association and organization. Our biochemical reconstitution system allowed for in-depth study of the biochemical interactions underlying the self-organization of Rab5 and its interacting molecules on the membrane.

# Results

## Upon GDP/GTP exchange Rab5 is directly transferred from Rabex5 to Rabaptin5 - a mechanistic basis for positive feedback of Rab5 activation

To directly test the positive feedback loop model, we investigated the structural rearrangements occurring in Rab5 and Rabex5/Rabaptin5 in the course of nucleotide exchange by Hydrogen Deuterium Exchange Mass Spectrometry (HDX-MS). Rabex5/Rabaptin5 was first premixed with Rab5:GDP and the resulting ternary complex diluted into deuterated buffer in the absence (*Figure 1A* top) or presence (*Figure 1A* bottom) of GTPγS and incubated for 1, 5 and 15 min. In this way, we could monitor structural rearrangement occurring in the early stages of the nucleotide exchange reaction. Focusing first on Rab5, we could see evidence of nucleotide exchange from the dramatic stabilization of Val24-Leu38, Leu130-Leu137 and Met160-Met168, encompassing the P-loop and parts of β 5, α4 and β 6 (*Figure 1B* and *Figure 1—figure supplement 1A*: dark blue), which, together, make up most of the direct interaction sites with GTP. In addition, we saw stabilization of Gln60-Phe71, parts of β 2 and β 3 (sky blue and pale green), consistent with the binding of Rabaptin5 (*Zhu et al., 2004*). Indication of binding to Rabaptin5 was observed after only 1 min of reaction, thus providing

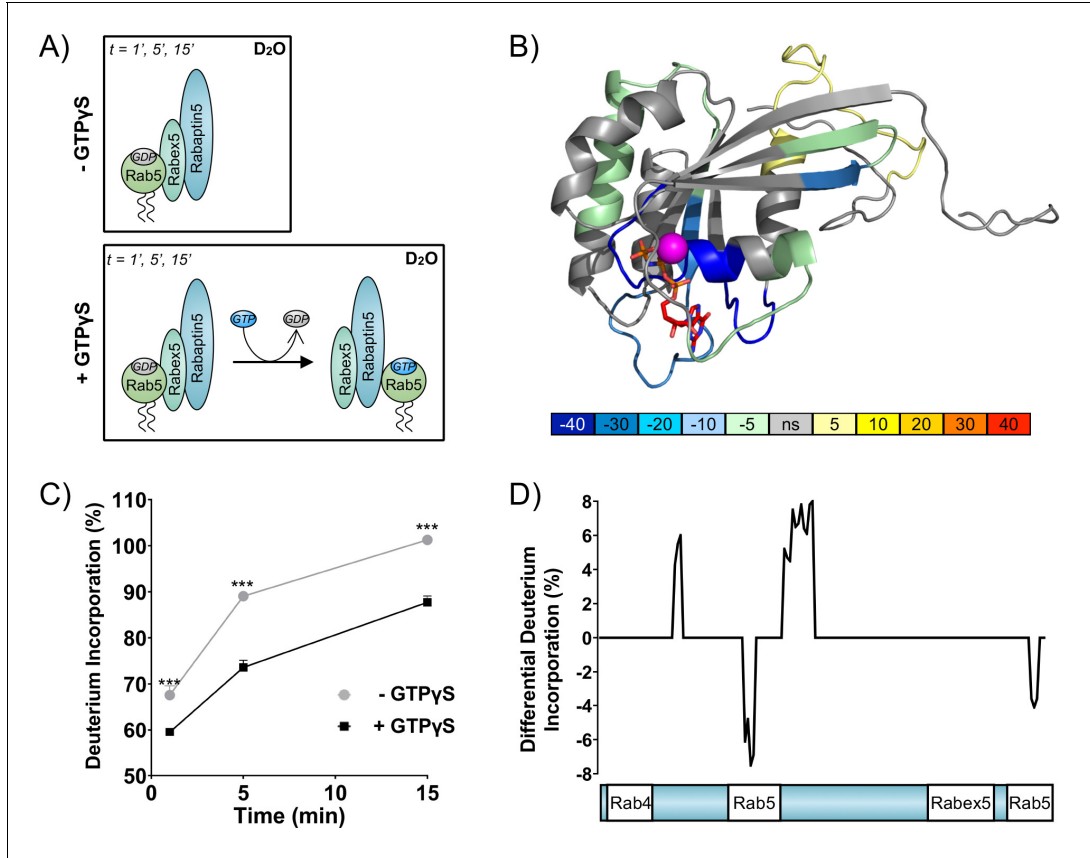

**Figure 1.** Rab5 backbone dynamics during nucleotide exchange. (A) Scheme of reaction. The ternary complex (Rab5/Rabex5/Rabaptin5) was incubated in $D_2O$ for 1, 5 or 15 min in the presence or absence of GTPγS. (B) Crystal structure of Rab5:GTP (PDBID: 3MJH) pseudocolored to show differential uptake of ternary complex (Rab5/Rabex5/Rabaptin5)±GTPγS (average of 1 min, 5 min and 15 min timepoints). The $Mg^{2+}$ ion is shown as a sphere (magenta) and GTPγS as a line structure. Color scheme: regions that are protected from exchange, i.e. stabilization, are colored with cool colors; regions with enhanced exchange with warm colors; regions with no statistically different uptake are colored in grey; and regions with no peptide coverage are white. (C) Deuterium incorporation over time in Rab5 β2 (aa 58–63, colored blue in B), in the ternary complex (Rab5/Rabex5/Rabaptin5)± GTPγS (n = 3) (D) Differential deuterium incorporation in Rabaptin5 during the nucleotide exchange reaction. Two areas of protection (decrease in deuterium uptake) correspond with the Rab5 binding sites.

The online version of this article includes the following figure supplement(s) for figure 1:

**Figure supplement 1.** Differential deuterium uptake of Rab5 and Rabaptin5 during nucleotide exchange.

evidence of a direct hand-off of active Rab5 from the Rabex5 catalytic domain to Rabaptin5 (See *Figure 1C*). Interestingly, we also saw a destabilization of Ile177-Asp200, α5 (yellow), suggesting a structural rearrangement of the C-terminal HVR. *Figure 1D* shows the alterations in deuterium exchange for Rabaptin5 (See also *Figure 1—figure supplement 1B*). Since there is no available structural model for Rabaptin5 the data are represented as a graph in which each peptide showing statistically significant alterations in deuterium uptake is assigned a value for the percent alteration. We saw stabilization in both of the regions known to bind Rab5, thus providing further evidence of Rab5 binding to Rabaptin5 after the nucleotide exchange reaction. This provides a putative structural mechanism for positive feedback loop formation and the need to couple GEF and effector activities. Next, we set out to test the hypothesis that such positive feedback is sufficient to induce the recruitment and localized accumulation of membrane-bound Rab5.

## Reconstituting Rab5 domain formation in vitro

To reconstitute Rab5 membrane recruitment and organization, we developed an in vitro system consisting of recombinant proteins and synthetic membranes. The lipid composition of the synthetic membrane was chosen based on the lipid composition of an enriched early endosomal fraction determined in a previous study (*Perini, 2012*). Briefly, an enriched early endosomal fraction was prepared as described (*Horiuchi et al., 1997*). Lipid extracts of this fraction were prepared in a final solution of $CHCl_3$:MeOH (1:2), as described by *Kalvodova et al., 2009*. The lipid extracts were then subjected to quantitative lipid analysis as described by *Sampaio et al., 2011*. The results are broadly in agreement with previous studies investigating the lipid composition of the plasma membrane which is known to be similar to that of the early endosome (*Casares et al., 2019*). Lipids constituting over 1 mol% of this lipid composition were utilized (EE, See *Table 1*). In order to test a wide number of experimental conditions, we designed the following workflow: small unilamellar vesicles with the EE-like lipid composition (EE-SUV) were deposited onto silica beads of 10 μm in diameter to form membrane-coated beads (EE-MCB). EE-MCBs were incubated with recombinant proteins, some of which were fluorescently tagged allowing us to monitor protein recruitment and spatial organization using confocal microscopy. EE-MCBs were segmented and visualized as Mollweide map projections. For visualization, the EE-MCBs are presented as equatorial slices in GFP/RFP and DiD channels, and the reconstructed bead surface as a Mollweide map projection of the GFP/RFP signal (Mollweide map projections of the DiD signal can be found in the corresponding supplementary figures).

EE-MCBs incubated with 10 nM GFP-Rab5/GDI showed membrane recruitment of GFP-Rab5 with a random distribution (see *Figure 2A* and *Figure 2—figure supplement 1A*). The addition of GDI and Rabex5/Rabaptin5-RFP in the presence GDP removed GFP-Rab5 from the membrane (see *Figure 2B* and *Figure 2—figure supplement 1B*). However, the same reaction in the presence of GTP produced a striking redistribution of GFP-Rab5 on the membrane into discrete clusters or domains (see *Figure 2C*, and *Figure 2—figure supplement 1C* and *Figure 2—Video 1*). GDI was provided in excess to allow for efficient flux of Rab5 between the membrane and soluble fraction

**Table 1.** Lipid compositions used in this study.

| | EE-MCB (mol %) | PC/PS/CH/PI(3)P/SM-MCB (mol%) | PC/PS/CH/PI(3)P/PlasmPE-MCB (mol%) | PC/PS/CH-MCB (mol%) | PC/PS-MCB (mol %) |
|---|---|---|---|---|---|
| Cholesterol | 32.2 | 32.2 | 32.2 | 32.2 | - |
| DOPC | 16.6/15.6 | 39.1 | 38.8 | 51.7 | 84.9/83.9 |
| Ethanolamine plasmalogen | 12.9 | - | 12.9 | - | - |
| Sphingomyelin | 12.6 | 12.6 | - | - | - |
| GM3 | 9 | - | - | - | - |
| DOPS | 6.1 | 15 | 15 | 15 | 15 |
| DOPE | 6.8 | - | - | - | - |
| Choline plasmalogen | 3.6 | - | - | - | - |
| PI(3)P | 0/1 | 1 | 1 | 0/1 | 0/1 |
| DiD | 0.1 | 0.1 | 0.1 | 0.1 | 0.1 |

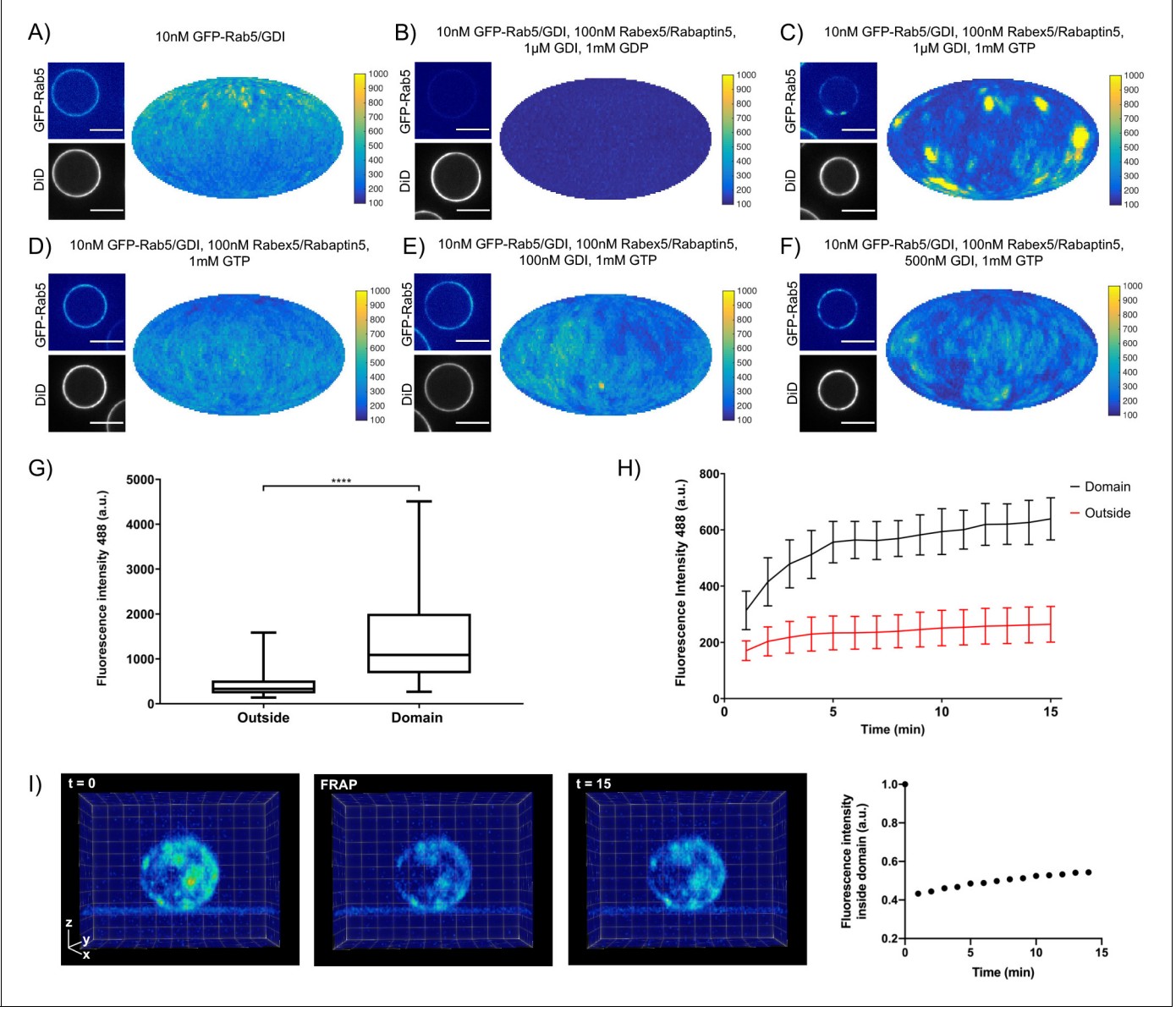

**Figure 2.** Rab5 domains can be reconstituted in vitro. EE MCBs were incubated for 15 min at 23 °C with 10 nM GFP-Rab5/GDI A) and supplemented with 1 µM GDI, 100 nM Rabex5/Rabaptin5-RFP and 1 mM GDP (B) or GTP (C). (D–F) GDI is necessary for Rab5 domain formation. EE MCBs were incubated with 10 nM GFP-Rab5/GDI complex, 100 nM Rabex5/Rabaptin5 1 mM GTP and 0 nM (D), 100 nM (E) or 500 nM (F) GDI. Beads are presented as equatorial slices in GFP and DiD channels (*left*) and a Mollweide projection of the GFP channel (*right*). Scale Bar = 10 µm. (G) Mean GFP-Rab5 signal intensity outside of and within segmented domains in C) (See also *Table 2*) (p=<0.0001) (H) EE MCBs were at 23 °C with 10 nM GFP-Rab5/GDI 1 µM GDI, 100 nM Rabex5/Rabaptin5 and 1 mM GTP and imaged in 1 min intervals for a total of 15 min. Graph presents mean GFP-Rab5 signal intensity outside of and within segmented domains over time (n = 63).). (I) EE MCBs were incubated for 15 min at 23 °C with 10 nM GFP-Rab5/GDI 1 µM GDI, 100 nM Rabex5/Rabaptin5 and 1 mM GTP (*panel 1*; ,t = 0') then bleached (*panel 2*; ,FRAP') and imaged in 1 min intervals for a total of 15 min. Shown here are stills from *Figure 2—Video 2* (*panels 1–3*) and average intensity within segmented domains over time (*panel 4*; n = 27).

The online version of this article includes the following video, source data, and figure supplement(s) for figure 2:

**Source data 1.** Rab5 domains can be reconstituted in vitro.
**Figure supplement 1.** Rab5 domains can be reconstituted in vitro.
**Figure 2—video 1.** Rab5 domains can be reconstituted in vitro.
https://elifesciences.org/articles/54434#fig2video1
**Figure 2—video 2.** Rab5 domains recover after photobleaching.
https://elifesciences.org/articles/54434#fig2video2

and, although catalytically active at much lower GTPase:GEF ratios, Rabex5/Rabaptin5 was provided in excess in order to allow for efficient binding between Rab5:GTP and Rabaptin5. Interestingly, the formation of GFP-Rab5 domains required GDI in a concentration-dependent manner (see *Figure 2D–F* and *Figure 2—figure supplement 1D-F*). GFP-Rab5 domains were segmented using Squassh (*Rizk et al., 2014*) on the surface of the bead (see *Figure 2—figure supplement 1G*), and the segmented structures were then characterized in terms of size and fluorescence intensity. *Table 2* summarizes the characteristics of GFP-Rab5 domains from experiments shown in *Figure 2*. Domains with a mean diameter of 1.32 µm were detected on MCBs incubated with GFP-Rab5/GDI, GDI, Rabex5/Rabaptin5, and GTP but not GDP. They formed with a characteristic density of ~4.7 domains/EE-MCB and were rarely found adjacent to one another. A critical hallmark of the reconstituted domains is a marked increase in GFP-Rab5 signal within the segmented domain as compared to the area outside (See *Figure 2G*). Comparison between GFP-Rab5 and DiD signals revealed that the occasional apparent clusters of GFP-Rab5 in the absence of other factors (see *Figure 2A* and *Figure 2—figure supplement 1A*) were due to membrane inhomogeneity characterized by lower DiD signal, unlike GFP-Rab5 domains.

In order to understand how these domains form, we monitored EE-MCBs over time (See *Figure 2H*, *Figure 2—Video 2*). Domains appear to be nucleated within the first minute of the reaction (which we could not capture due to the imaging setup) and then grow linearly in intensity until ~5 min after initiation of the reaction. After this point individual domains increase in GFP-Rab5 signal intensity slowly or not at all, suggesting that some domains reach saturation. Interestingly, when whole MCBs were bleached domains recovered in the same locations after photobleaching indicating that there is a constant exchange of GFP-Rab5 with solution (See *Figure 2I*, *Figure 2—Video 2*).

## Rabex5/Rabaptin5 is essential for Rab5 domain formation in vitro

In order to understand the mechanisms by which Rab5 domains form, we dissected the contribution of each component of our reconstituted system. GDI delivers and extracts Rab5, as seen in *Figure 2*, and is essential for domain formation. We observed that, similar to GDI, Rab5 domain formation requires Rabex5/Rabaptin5 in a concentration-dependent manner (see *Figure 3A–E*, *Figure 3—figure supplement 1A-E* and *Table 3*, which summarizes the conditions shown in *Figure 3A,B & C*).

Next, we verified that the Rabex5/Rabaptin5 complex indeed localizes to the Rab5 domain. For this, we used a fluorescent Rabex5/Rabaptin5-RFP complex and observed both enrichment of Rabaptin5-RFP signal inside the domain (See *Figure 4A*) and colocalization with GFP-Rab5 (See *Figure 4B*). Rabex5/Rabaptin5-RFP also showed some degree of membrane association in the

**Table 2.** Rab5 domains can be reconstituted in vitro.
EE MCBs were incubated for 15 min at 23 ℃ with 10 nM GFP-Rab5/GDI and supplemented with 1 µM GDI, 100 nM Rabex5/Rabaptin5-RFP and 1 mM GDP or GTP.

|  | 10 nM GFP-Rab5/GDI | 10 nM GFP-Rab5/GDI, 100 nM Rabex5/Rabaptin5, 1 µM GDI, 1 mM GDP | 10 nM GFP-Rab5/GDI, 100 nM Rabex5/Rabaptin5, 1 µM GDI, 1 mM GTP |
|---|---|---|---|
| # Domains | 0 | 0 | 449 |
| # Beads | 30 | 44 | 96 |
| Mean # Domains/Bead | 0 | 0 | 4.7 |
| Mean intensity/Bead (a.u.) | 212.31 ± 67.04 | 128.40 ± 4.91 | 447.96 ± 403.41 |
| Mean Standard Deviation/Bead | 46.70 ± 21.26 | 0.36 ± 0.04 | 237.24 ± 225.54 |
| Mean Intensity/Domain | - | - | 1326.95 ± 1026.96 |
| Mean Intensity/Outside | 212.31 ± 67.04 | 128.40 ± 4.91 | 454.63 ± 364.79 |
| Mean domain area, µm$^2$ | - | - | $1.74^{+4.74}_{-1.00}$ |
| Mean domain diameter, µm | - | - | 1.32 |

The online version of this article includes the following source data for Table 2:
Source data 1. Rab5 domains can be reconstituted in vitro.

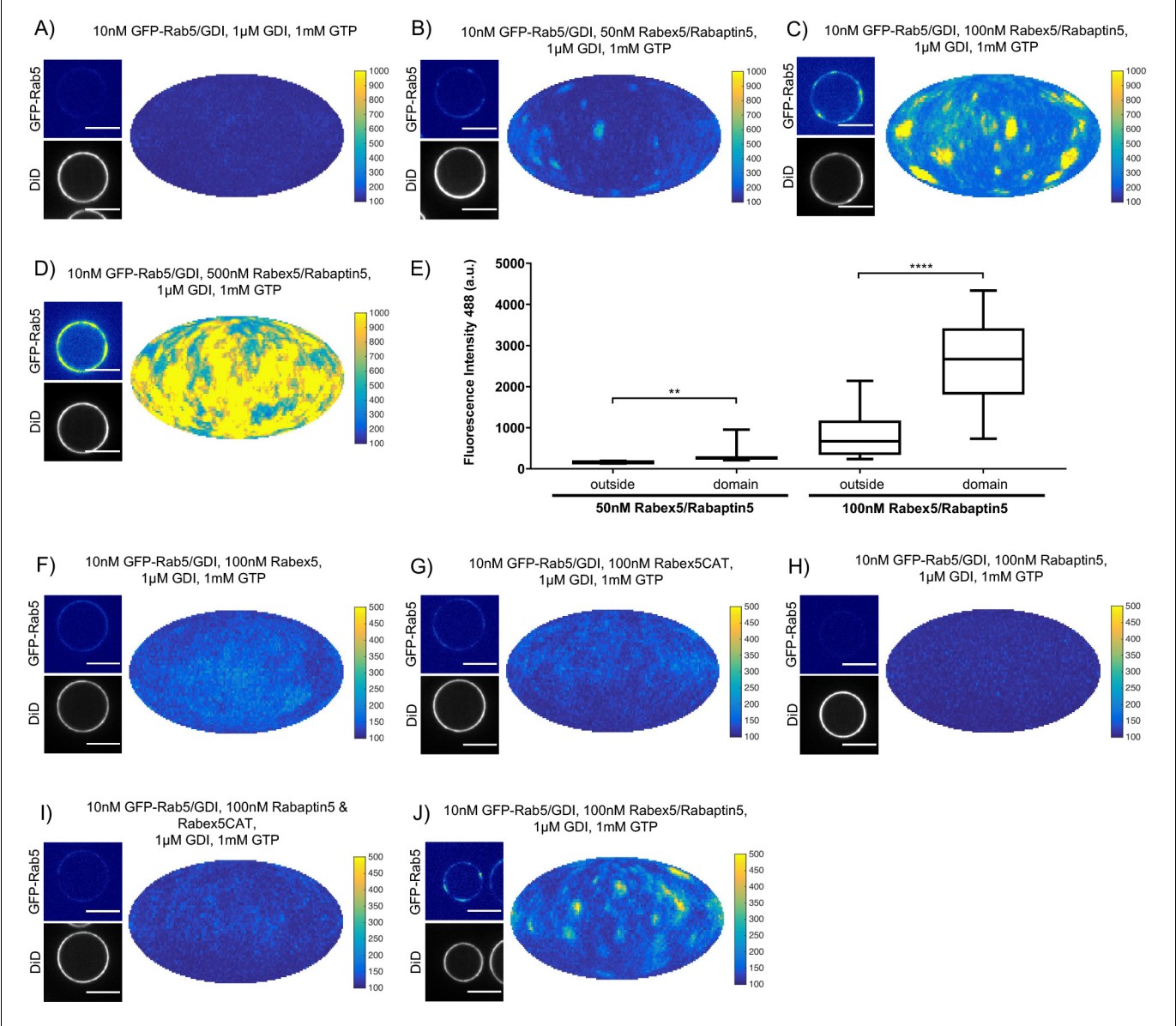

**Figure 3.** Rabex5/Rabaptin5 is essential for Rab5 domain formation in vitro. (**A - E**) Domain formation is dependent on concentration of Rabex5/Rabaptin5. EE MCBs were incubated for 15 min at 23 °C with 10 nM GFP-Rab5/GDI, 1 μM GDI, 1 mM GTP and 0 nM (**A**), 50 nM (**B**), 100 nM (**C**), or 500 nM (**D**) Rabex5/Rabaptin5-RFP. (**E**) Mean GFP-Rab5 signal intensity outside of and within segmented domains as a function of Rabex5/Rabaptin5 concentration (50 nM Rabex5/Rabaptin5 p=0.001, n = 95; 100 nM Rabex5/Rabaptin5 p=<0.0001) See also **Table 3**) (**F – J**) Rabex5/Rabaptin5 cannot be split into component parts and still form domains. EE MCBs were incubated for 15 min at 23 °C with 10 nM GFP-Rab5/GDI, 1 μM GDI, 1 mM GTP and 100 nM Rabex (**F**), 100 nM RabexCAT (**G**), 100 nM Rabaptin5 (**H**), 100 nM Rabex5CAT and Rabaptin5 (**I**), or 100 nM Rabex5/Rabaptin5 (**J**). Beads are presented as equatorial slices in GFP and DiD channels (*left*) and a Mollweide projection of the GFP channel (*right*). Scale Bar = 10 μm.

The online version of this article includes the following figure supplement(s) for figure 3:

**Figure supplement 1.** Rabex5/Rabaptin5 is essential for Rab5 domain formation in vitro.

absence of other factors (See **Figure 3—figure supplement 1F**), however this was significantly lower than the signal observed inside the GFP-Rab5 domains.

We next wanted to investigate whether the full Rabex5/Rabaptin5 complex was necessary for domain formation (**Figure 3F–J** and **Figure 3—figure supplement 1G-K**). In the presence of Rab5, GDI and GTP, neither full-length Rabex5 nor the Rabex5 catalytic domain (Rabex5CAT) alone were sufficient to form domains (**Figure 3F,G**). Similarly, Rabaptin5 alone was not capable of forming

**Table 3.** Domain formation is dependent on concentration of Rabex5/Rabaptin5.
EE MCBs were incubated for 15 min at 23 °C with 10 nM GFP-Rab5/GDI, 1 μM GDI, 1 mM GTP and 0 nM, 50 nM, 100 nM Rabex5/Rabaptin5-RFP. Beads incubated with 10 nM GFP-Rab5/GDI, 1 μM GDI, 1 mM GTP and 500 nM Rabex5/Rabaptin5-RFP could not be properly segmented due to the high GFP-Rab5 signal on the bead (See *Figure 3D*).

| | 10 nM GFP-Rab5/GDI, 1 μM GDI, 1 mM GTP | 10 nM GFP-Rab5/GDI, 50 nM Rabex5/Rabaptin5, 1 μM GDI, 1 mM GTP | 10 nM GFP-Rab5/GDI, 100 nM Rabex5/Rabaptin5, 1 μM GDI, 1 mM GTP |
|---|---|---|---|
| # Domains | 0 | 96 | 90 |
| # Beads | 17 | 23 | 16 |
| Mean # Domains/Bead | 0 | 4.17 | 5.63 |
| Mean intensity/Bead (a.u.) | 132.95 ± 6.23 | 164.66 ± 24.13 | 946.76 ± 669.27 |
| Mean Standard Deviation/Bead | 13.14 ± 2.68 | 41.63 ± 17.87 | 526.77 ± 332.23 |
| Mean Intensity/Domain | - | 282.58 ± 96.68 | 2767.14 ± 1039.34 |
| Mean Intensity/Outside | 132.95 ± 6.23 | 159.56 ± 18.33 | 856.22 ± 573.11 |
| Mean domain area, μm$^2$ | - | $1.71^{+3.36}_{-0.95}$ | $1.97^{+6.22}_{-1.26}$ |
| Mean domain diameter, μm | - | 1.31 | 1.40 |

The online version of this article includes the following source data for Table 3:
**Source data 1.** Domain formation is dependent on concentration of Rabex5/Rabaptin5.

domains (*Figure 3H*). Unlike the full length Rabex5/Rabaptin5 complex, Rabex5CAT plus full-length Rabaptin5 did not support Rab5 domain formation (compare *Figure 3J and I*). This suggests that direct coupling of GEF activity and effector binding is essential for Rab5 domain formation.

Finally, we quantified the domain size distribution as a function of concentration of the components in the reaction. Interestingly, neither the domain diameter nor the area differed significantly

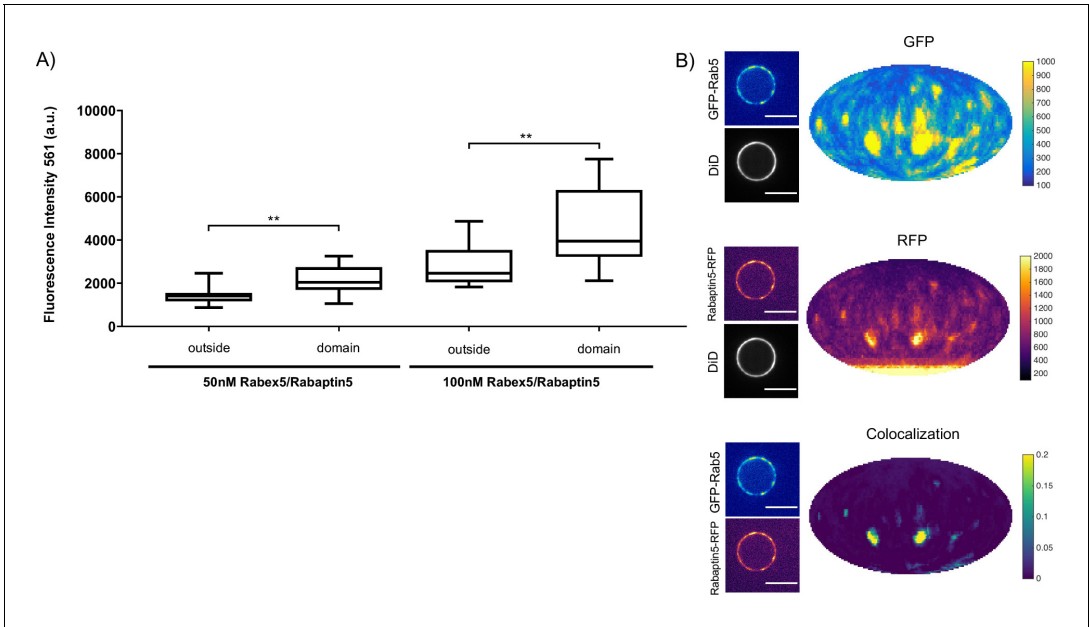

**Figure 4.** Rabex5/Rabaptin5 localises to the reconstituted Rab5 domain. EE MCBs were incubated for 15 min at 23 °C with 10 nM GFP-Rab5/GDI, 1 μM GDI, 1 mM GTP and 50 nM or 100 nM Rabex5/Rabaptin5-RFP (See *Figure 3A–E*). (A) Rabaptin5-RFP signal is enriched in domains. (50 nM Rabaex5/Rabaptin5 p=0.001, n = 96; 100 nM Rabex5/Rabaptin5 p=0.0017, n = 90). Corresponding GFP enrichment in presented in *Figure 3E*. (B) Equatorial slices and mollweide representations of GFP signal (*top*), RFP signal (*bottom*) and pixelwise GFP-RFP colocalization (*bottom*). Beads are presented as equatorial slices (*left*) and Mollweide projections (*right*). Scale Bar = 10 μm.

when decreasing Rabex5/Rabaptin5 concentration, but the mean intensity of domains decreased with decreasing concentration of Rabex5/Rabaptin5 (See *Figure 3E* and *Table 3*).

## Rab5 domain formation is influenced by membrane composition

In addition to protein-protein interactions, protein-lipid and lipid-lipid interactions also play a role in Rab5 domain formation. The above experiments (*Figures 2* and *3*) were all conducted with the EE lipid composition containing 1 mol% PI(3)P. The rearrangements in Rab5 during nucleotide exchange reveal a destabilization of α5 that may alter membrane contacts or orientation of the protein with respect to the membrane in the GDP- vs GTP-bound conformation (See *Figure 1B*). Previous work using molecular dynamics simulations suggested an interaction between the Rab5 HVR and PI(3)P as well as cholesterol (*Edler and Stein, 2017b*). To investigate the contribution of lipids, specifically PI(3)P and cholesterol, to GFP-Rab5 domain formation, EE-MCBs as well as MCBs with a simple PC/PS lipid composition were made with either 1 mol% or 0 mol% PI(3)P (PC/PS-MCB; See *Table 1*). Geranylgeranylated GFP-Rab5 was recruited similarly to EE-MCBs and PC/PS-MCBs that included 1 mol% PI(3)P (see *Figure 5A,B E* and *Figure 5—figure supplement 1A B*). However, recruitment of GFP-Rab5 to both membranes lacking PI(3)P was greatly diminished (see *Figure 5C–E* and *Figure 5—figure supplement 1C D*). This suggests that the presence of PI(3)P enhances Rab5 recruitment, either by facilitating the dissociation of Rab5 from GDI or by inhibiting the extraction of Rab5 by GDI. The presence of cholesterol (CH) appeared to also improve Rab5 recruitment to the simple lipid composition, although to a lesser degree than PI(3)P (See *Figure 5F*; PC/PS/CH-MCB vs PC/PS-MCB, See *Table 1*). Similar investigation of the contribution of cholesterol in the EE-like lipid composition was not possible in this system as membrane integrity was greatly compromised without cholesterol (data not shown).

In order to determine whether these interactions have an effect on domain formation, the same MCBs were incubated with Rab5/GDI, Rabex5/Rabaptin5, GDI and GTP. Strikingly, domain formation was most efficient on EE-MCBs with 1 mol% PI(3)P, less efficient on EE-MCBs with 0 mol% PI(3)P and completely abolished on PC/PS membranes regardless of PI(3)P content (see *Figure 6*, *Figure 6—figure supplement 1* and *Table 4* which summarizes the conditions shown in *Figure 6*). Domains formed on EE membranes in the absence of PI(3)P had a drastically reduced mean domain intensity (508.32 ± 143.37) compared to domains formed in the presence of PI(3)P (mean domain intensity 1269.32 ± 556.54) (See *Figure 6E*). Importantly, the membrane association of Rabex5/Rabaptin5-RFP was not found to be similarly lipid composition-dependent (See *Figure 3—figure supplement 1F*).

To investigate further which components of the EE lipid composition contribute to domain formation, the simple PC/PS lipid composition was sequentially modified to include the three most abundant lipids in the EE lipid composition: cholesterol (32.3 mol%), sphingomyelin (SM; 12.6 mol%) and ethanolamine plasmalogen (PlasmPE; 12.9 mol%) (See *Figure 6F*; *Table 5*). PI(3)P was included due to the aforementioned effect on Rab5 recruitment. Interestingly, no domain formation could be observed for PC/PS-MCBs while both PC/PS/CH-MCBs and PC/PS/CH/PlasmPE-MCBs showed infrequent and only low-intensity domains compared to EE-MCBs. Only PC/PS/CH/SM-MCBs were able to recapitulate domains with a GFP signal intensity similar to that observed on EE-MCBs, although the prevalence of domains on PC/PS/CH/SM-MCBs was still reduced. Staining EE-MCBs with C-laurdan in the absence of proteins and imaging by confocal microscopy, as described by *Dodes Traian et al., 2012* did not reveal the presence of macroscopic pre-existing liquid ordered (L$_O$) domains (data not shown).

The observation that Rab5 can be recruited efficiently to PC/PS/PI(3)P membranes but cannot be organized into domains in the presence of Rabex5/Rabaptin5, excess GDI, and GTP suggest that Rab5 interacts differently with the complex EE membrane that with a simple PC/PS membrane. Our results demonstrate that PI(3)P enhances recruitment of Rab5 to MCBs and the presence of lipids that contribute to acyl chain packing (cholesterol, sphingomyelin; *Kaiser et al., 2009*) are necessary to drive Rab5 domain formation.

## Discussion

GEF/effector coupling and the resulting positive feedback loop of GTPase activation and membrane recruitment are common to many small GTPase systems and have been implicated in their spatial

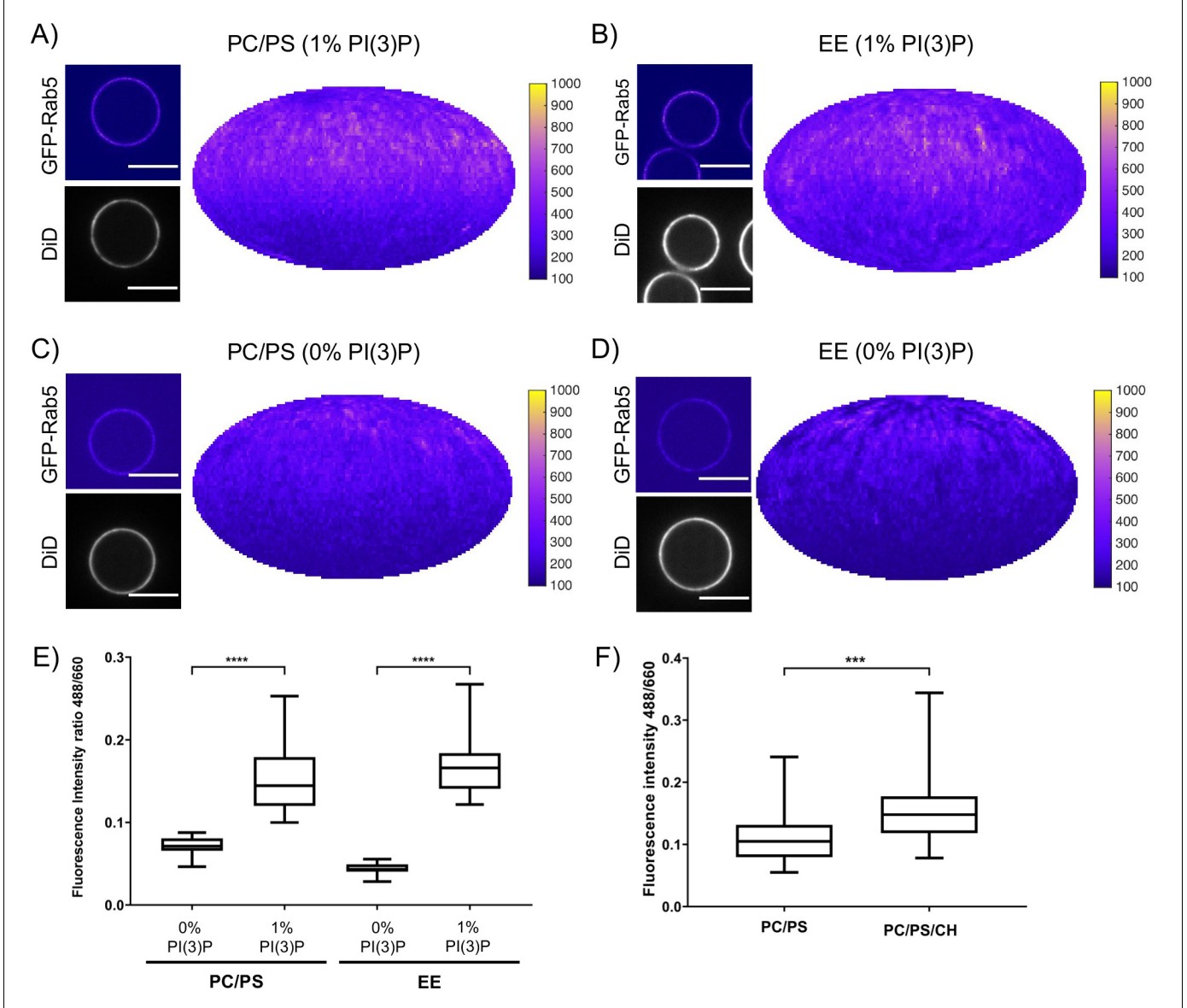

**Figure 5.** Recruitment of geranylgeranylated GFP-Rab5 to EE and PC/PS bilayers is enhanced by PI(3)P. MCBs with PC/PS and EE lipid composition containing 1 mol% PI(3)P (**A**) and **B**) respectively) and MCBs with PC/PS and EE lipid composition containing 0 mol% PI(3)P (**C**) and **D**) respectively) were incubated with 10 nM GFP-Rab5/GDI for 15 min at 23 ˚C. Beads are presented as equatorial slices in GFP and DiD channels (*left*) and Mollweide projection of the GFP channel (*right*). Scale Bar = 10 µm. (**E**) Mean equatorial GFP signal intensity in **A–D**). (p=<0.0001; n = 20) (**F**) MCBs with PC/PS and PC/PS/CH lipid composition (0 mol% PI(3)P) incubated with 10 nM GFP-Rab5/GDI for 15 min at 23 ˚C. Graph presents mean equatorial GFP signal intensity (p=0.005; n = 25). For both **E**) and **F**) GFP signal intensity is normalized to DiD signal intensity, however the same pattern can be seen in the raw intensity values.

The online version of this article includes the following figure supplement(s) for figure 5:

**Figure supplement 1.** Recruitment of geranygeranylated GFP-Rab5 to EE and PC/PS bilayers is enhanced by PI(3)P.

patterning. In this study, we demonstrated that membrane recruitment and extraction (via GDI) together with coupling of GEF and effector activities (via Rabex5/Rabaptin5) are sufficient to reconstitute domain organization of Rab5 in vitro. Geranylgeranylated Rab5 was observed to be recruited to EE-like membranes from the Rab5/GDI complex. Whereas in the absence of other factors Rab5 was randomly distributed in the plane of the membrane, upon the addition of GDI, Rabex5/Rabaptin5 and GTP, it reorganized into discrete domains in a GTP-dependent manner. Key to Rab5 domain formation was the 'handover' of Rab5 from Rabex5 to Rabaptin5 and the lipid composition

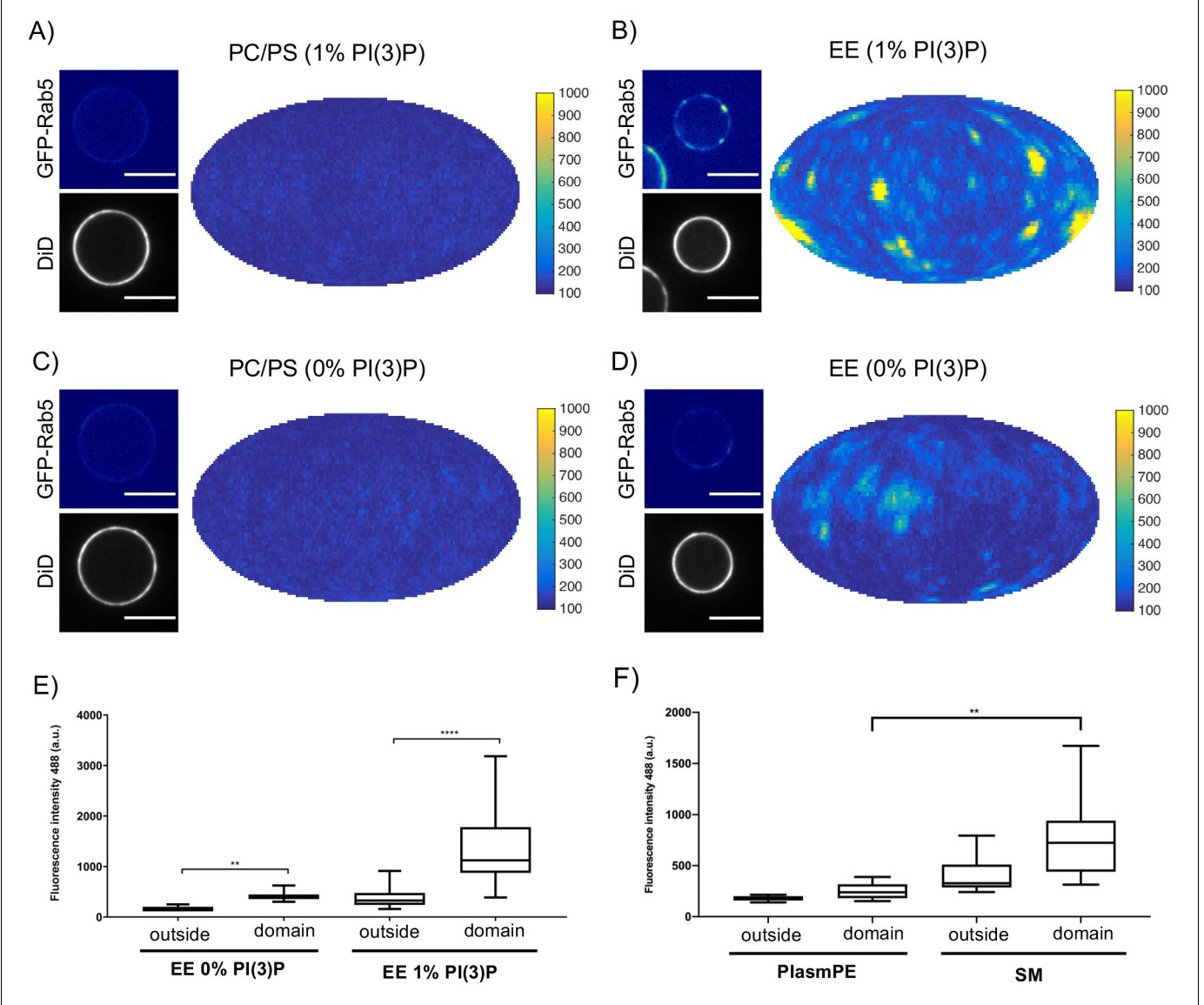

**Figure 6.** Rab5 domain formation in vitro is influenced by membrane composition. MCBs with PC/PS and EE lipid composition containing 1 mol% PI(3)P (**A**) and **B**) respectively) and MCBs with PC/PS and EE lipid composition containing 0 mol% PI(3)P (**C**) and **D**) respectively) were incubated with 10 nM GFP-Rab5/GDI, 1 μM GDI, 100 nM Rabex5/Rabaptin5-RFP and 1 mM GTP for 15 min at 23 ˚C. Beads are presented as equatorial slices in GFP and DiD channels (*left*) and Mollweide projection of the GFP channel (*right*). Scale Bar = 10 μm. (**E**) Mean GFP-Rab5 signal intensity outside of and within segmented domains in **B**) and **D**) (p=<0.0001) (See also *Table 2*). (**F**) Mean GFP-Rab5 signal intensity outside of and within segmented domains on MCBs with PC/PS/CH/PlasmPE and PC/PS/CH/SM lipid composition containing 1 mol% PI(3)P (p=<0.0046) (See also *Table 5*).

The online version of this article includes the following figure supplement(s) for figure 6:

**Figure supplement 1.** Rab5 domain formation in vitro is influenced by membrane composition.

of early endosomes, suggesting a hitherto unknown cooperativity between lipids and Rab-dependent membrane self-organization.

Self-organizing systems that form spatial patterns on membranes often depend on non-linear dynamics (*Halatek et al., 2018*). In our system, a key feature is the membrane recruitment and activation of Rab5, regulated by the Rabex5/Rabaptin5 complex. Neither GEF activity nor effector binding alone were capable of supporting domain formation unless physically coupled in a complex. We found that, in the course of nucleotide exchange, newly activated Rab5 is released from Rabex5 and immediately binds Rabaptin5 suggesting there is a direct delivery or 'handover' of Rab5 from

**Table 4.** Rab5 domain formation in vitro is influenced by membrane composition.

MCBs with EE and PC/PS lipid composition containing 1 mol% PI(3)P and MCBs with EE and PC/PS lipid composition containing 0 mol% PI(3)P were incubated with 10 nM GFP-Rab5/GDI, 1 µM GDI, 100 nM Rabex5/Rabaptin5-RFP and 1 mM GTP for 15 min at 23 °C.

| | PC/PS (0% PI(3)P) | PC/PS (1% PI(3)P) | EE (0% PI(3)P) | EE (1% PI(3)P) |
|---|---|---|---|---|
| # Domains | 0 | 0 | 13 | 164 |
| # Beads | 33 | 38 | 24 | 40 |
| Mean # Domains/Bead | 0 | 0 | 0.54 | 4.1 |
| Mean intensity/Bead (a.u.) | 135.48 ± 14.69 | 129.54 ± 11.79 | 140.88 ± 39.73 | 429.23 ± 217.66 |
| Mean Standard Deviation/Bead | 16.69 ± 8.60 | 13.23 ± 6.02 | 26.05 ± 25.50 | 245.40 ± 120.62 |
| Mean Intensity/Domain | - | - | 508.32 ± 143.37 | 1269.32 ± 556.54 |
| Mean Intensity/Outside | 135.48 ± 14.69 | 129.54 ± 11.79 | 138.59 ± 32.05 | 393.35 ± 194.66 |
| Mean domain area, µm$^2$ | - | - | $2.12^{+4.76}_{-1.21}$ | $1.42^{+2.67}_{-0.73}$ |
| Mean domain diameter, µm | - | - | 1.46 | 1.19 |

The online version of this article includes the following source data for Table 4:

Source data 1. Rab5 domain formation in vitro is influenced by membrane composition.

Rabex5 to Rabaptin5. This 'handover' is likely facilitated by the dimerization of the Rabex5/Rabaptin5 complex and presents a structural mechanism by which a positive feedback loop of Rab5 activation could be generated. Other Rab5 GEFs that localize and recruit Rab5 to different intracellular compartments (e.g. GAPVD1 or RIN1 on clathrin-coated vesicles and the plasma membrane; *Tall et al., 2001*; *Semerdjieva et al., 2008* have as of yet not been found to be coupled to effector activity. In vivo Rabex5/Rabaptin5 can be targeted to the EE by interaction of Rabex5 with ubiquitinated receptors and the binding of Ubiquitin to Rabex5 enhances nucleotide exchange activity (*Lee et al., 2006*; *Mattera et al., 2006*; *Penengo et al., 2006*; *Lauer et al., 2019*). This implies that ubiquitinated cargo can act not only to recruit Rabex5/Rabaptin5 but also potentially contribute to Rab5 domain formation and/or localization on the EE.

The membrane diffusion of multiple small GTPases has been shown to be integral in their capacity to self-organize (*Goryachev and Pokhilko, 2008*; *Bruurs et al., 2015* and *Bruurs et al., 2017*). An important new finding of this study is the role of lipids in Rab5 domain formation. In our reconstituted system, PI(3)P and cholesterol enhanced the membrane recruitment of Rab5. In molecular

**Table 5.** Acyl chain ordering influences Rab5 domain formation.

MCBs with EE, PC/PS, PC/PS/CH, PC/PS/CH/PlasmPE and PC/PS/CH/SM lipid composition, each containing 1 mol% PI(3)P, were incubated with 10 nM GFP-Rab5/GDI, 1 µM GDI, 100 nM Rabex5/Rabaptin5-RFP and 1 mM GTP for 15 min at 23 °C.

| | PC/PS (1% PI(3)P) | PC/PS/CH (1% PI(3)P) | PC/PS/CH/PlasmPE (1% PI(3)P) | PC/PS/CH/SM (1% PI(3)P) | EE (1% PI(3)P) |
|---|---|---|---|---|---|
| # Domains | 0 | 80 | 78 | 87 | 163 |
| # Beads | 18 | 30 | 21 | 25 | 32 |
| Mean # Domains/Bead | 0 | 2.67 | 3.71 | 3.48 | 5.09 |
| Mean intensity/Bead (a.u.) | 144.70±20.92 | 171.03±64.72 | 181.34±79.51 | 301.41±175.91 | 525.67±181.34 |
| Mean Standard Deviation/Bead | 18.88±8.45 | 44.18±41.94 | 50.59±35.40 | 146.88±89.56 | 189.43±63.05 |
| Mean Intensity/Domain | - | 303.44±129.53 | 381.67±178.58 | 743.88±400.00 | 830.66±323.40 |
| Mean Intensity/Outside | 144.70±20.92 | 163.96±53.54 | 139.62±111.45 | 265.97±170.82 | 512.97±181.20 |
| Mean domain area, µm$^2$ | - | $2.45^{+5.35}_{-1.54}$ | $2.26^{+4.20}_{-1.33}$ | $2.09^{+5.00}_{-1.23}$ | $2.52^{+3.43}_{-1.55}$ |
| Mean domain diameter, µm | - | 1.57 | 1.50 | 1.45 | 1.59 |

The online version of this article includes the following source data for Table 5:

Source data 1. Acyl chain ordering influences Rab5 domain formation.

dynamics simulations, *Edler et al., 2017a* suggest a direct interaction between Rab5 and PI(3)P and also observed accumulation of cholesterol in the proximity of Rab5. The requirement for PI(3)P has important implications for the in vivo formation of Rab5 domains. On the EE, PI(3)P is mainly produced by the activity of the class II PI3K complex, Vps34/Vps15, which is regulated by a direct interaction between Rab5 and Vps15 (*Christoforidis et al., 1999a*; *Christoforidis et al., 1999b*; *Murray et al., 2002*; *Falasca and Maffucci, 2012*). This suggests that Rab5 directly modifies the local lipid environment to stabilize itself on the membrane, thus providing yet another level of positive feedback in vivo.

Similar to Rab5 recruitment, domain formation was lipid composition dependent and most strongly observed on membranes containing the full EE lipid mixture. The observation that simple, highly diffusive, PC/PS membranes do not support domain formation suggests that the EE lipid composition facilitates lateral lipid packing and protein-lipid interactions that are necessary for domain formation. *Lebrand et al., 2002* reported that cholesterol regulates the membrane association and activity of Rab7 on late endosomes in vivo and decreases GDI extraction of Rab7 in vitro. The requirement of cholesterol for stabilizing Rab5 on the membrane provides support to the idea that lipid packing serves to adapt to the longer chain length of the geranylgeranyl anchor. Increasing the complexity of the PC/PS lipid composition by adding cholesterol allowed for the formation of few domains of low GFP signal intensity. The addition of sphingomyelin but not ethanolamine plasmalogen was sufficient to produce domains with a high GFP-Rab5 signal intensity, as observed with the EE lipid composition. Other than being the next most abundant lipids (after cholesterol) in the EE lipid composition, both sphingomyelin and ethanolamine plasmalogen increase the rigidity of the membrane. This increased rigidity occurs via the saturated acyl chains of sphingomyelin and the small headgroup of the ethanolamine plasmalogen, two very different mechanisms that both reduce diffusivity of the membrane. The observation that sphingomyelin was necessary for domain formation with intensity values comparable to the EE lipid mixture, but ethanolamine plasmalogen was not, suggests that it is not a reduction in global membrane diffusivity that enables domain formation, but the presence of saturated acyl chains and capacity for lateral lipid packing (*Kaiser et al., 2009*). This may allow for dense packing of geranylgeranyl chains and stabilize a nascent domain by locally reducing the diffusion of Rab5. Further, the destabilization of $\alpha 5$, which extends into the HVR, observed in Rab5 by HDX-MS may alter the conformation of membrane-bound Rab5 upon nucleotide exchange. In molecular dynamics simulations, *Edler and Stein, 2017b* observed a rotation within the membrane of Rab5:GTP with respect to Rab5:GDP, that not only exposes the effector binding site but also suggests that Rab5 makes different membrane contacts depending on its nucleotide state. Further molecular dynamics simulations showed that this nucleotide state-dependent orientation, as well as correct insertion of the geranylgeranyl anchors into the lipid bilayer, is only supported by an EE-like membrane, containing PI(3)P, cholesterol, sphingomyelin and charged lipids (*Edler and Stein, 2017b*; *Münzberg and Stein, 2019*). However, the EE-lipid composition alone did not yield macroscopic $L_O$ domains. Rab5 domain formation therefore requires the synergy between the Rab5 minimal machinery and lateral lipid packing. We suggest that the EE lipid composition supports Rab5 domain formation in our in vitro system through a combination of 1) direct interactions between Rab5 and PI(3)P enhancing recruitment to the membrane, 2) cholesterol stabilizing the geranylgeranyl anchor insertions to support a nucleotide-dependent orientation of Rab5 relative to the membrane, and 3) the presence of saturated lipids allowing for dense packing of geranylgeranyl chains, contributing to domain stabilization and growth.

The non-linearity of the nucleotide cycle coupled to specific lipid interactions make small GTPases widespread regulators of membrane self-organization. K-Ras for example, has long been known to cluster and alter the local lipid environment, e.g. by forming nanoclusters of PI(4,5)P$_2$ on the PM (*Zhou et al., 2017*). However, with the same design, different GTPase systems can form one (e.g. Cdc42) or multiple domains (e.g. ROP11, Rab5). Our in vitro system recapitulates the formation of multiple Rab5 domains on the same membrane. In the reconstituted system, Rab5 domains were formed with a characteristic density of ~4.7 domains/EE-MCB surface and a mean area of 1.74 $\mu m^2$ (which would be estimated to contain up to ~10,000 molecules of Rab5). *Chiou et al., 2018* propose that coexistence of multiple GTPase domains can arise if the density of active GTPase in the domain reaches a 'saturation' point. This would slow competition between domains, allowing multiple domains to exist simultaneously, and could occur via multiple biologically relevant mechanisms (e.g. local depletion of components or strong negative feedback). In our reconstituted system, we indeed

saw both characteristic spacing of domains and saturation of GFP-Rab5 signal, indicating that such a 'saturation' point can be reached. From the domain intensity we could observe two phases in domain growth, an initial phase characterized by rapid increase in GFP signal intensity over time, and a second phase characterized by slow increase or even saturation in signal intensity. We suggest that fast growth is dominated by reorganization of the local lipid environment and rapid recruitment of proteins from solution. Upon depletion of the critical components from the local membrane, domains stabilize and reach a second, slow-growing or saturated phase. We suggest that in this phase, domains reach dynamic equilibrium where domain size has stabilized but the domain continues to exchange proteins with the soluble pool, as suggested by the observation that domains recover in the same location after photobleaching. It may therefore be the interaction with the lipid membrane that stabilizes and determines the size of the domains obtained in our system. Further, it is apparent during purification that recombinant Rab5 dimerizes at high concentrations and this dimerization is enhanced by geranylgeranylation (data not shown). Given that domains create a locally high concentration of protein, Rab5 dimerization may also contribute to stabilization of a Rab5 domain. How domain growth is regulated and by what means biological systems can produce a variety of spatial patterns based on common design principles has been the subject of multiple recent in silico models and simulations (*Chiou et al., 2018*; *Halatek et al., 2018*; *Jacobs et al., 2019*). For Turing-type reaction-diffusion systems, theory can predict regions in the parameter space where the system either can form dynamically stable patterns or support oscillatory behavior such as travelling waves. In our reconstituted system, we did not observe such oscillatory behavior. This may either be a limitation of the experimental set-up (e.g. too low resolution in time and space) or an intrinsic property of the system we investigated. The design principles of biochemical oscillators require delayed negative feedback (*Novák and Tyson, 2008*) which might not be achieved by the linear GTP to GDP hydrolysis and GDI extraction reconstituted in our reactions. Further adaptation of the system (e.g. the addition of GAP activity) may allow for wave-like behavior and remains for future investigation. Our results imply that the specific interaction of proteins with lipids in the membrane must also be considered in in silico studies of pattern formation.

Herein, we reconstituted a minimal system for the formation of Rab5 GTPase domains in vitro and demonstrated that both GEF/effector coupling and lipid interactions contribute to the self-organization of Rab5 on the membrane, where the lipid composition plays an important role beyond that of a solvent for lipidated proteins. This appears to be a universal system deploying small GTPases to pattern membranes from mono-cellular to multi-cellular organisms.

## Materials and methods

**Key resources table**

| Reagent type (species) or resource | Designation | Source or reference | Identifiers | Additional information |
|---|---|---|---|---|
| Recombinant DNA reagent | pOEM-1 N-His | Oxford Expression Technologies, MPI-CBG PEP facility | | vector, NotI and AscI sites used for ligation |
| Recombinant DNA reagent | pOEM-1 N-GST | Oxford Expression Technologies, MPI-CBG PEP facility | | vector, NotI and AscI sites used for ligation |
| Recombinant DNA reagent | pOEM-1 N-His-eGFP | Oxford Expression Technologies, MPI-CBG PEP facility | | vector, NotI and AscI sites used for ligation |
| Recombinant DNA reagent | pOEM-1 C-His-tagRFP | Oxford Expression Technologies, MPI-CBG PEP facility | | vector, NotI and AscI sites used for ligation |
| Transfected construct (*Homo sapiens*) | Rab5a | This paper | | In vector pOEM-1 N-His-eGFP |
| Transfected construct (*Bos taurus*) | Rabex5 (pOEM-1 N-His) | *Lauer et al., 2019* | | In vector pOEM-1 N-His |

*Continued on next page*

*Continued*

| Reagent type (species) or resource | Designation | Source or reference | Identifiers | Additional information |
|---|---|---|---|---|
| Transfected construct (*Bos taurus*) | Rabex5CAT (pOEM-1 N-His) | *Lauer et al., 2019* | | In vector pOEM-1 N-His |
| Transfected construct (*Homo sapiens*) | Rabaptin5 (pOEM-1 N-GST) | *Lauer et al., 2019* | | In vector pOEM-1 N-GST |
| Transfected construct (*Homo sapiens*) | Rabaptin5-RFP-6xHis | This paper | | pOEM-1 C-His-tagRFP |
| Transfected construct (*Homo sapiens*) | GDIA (pOEM-1 N-His) | This paper | | In vector pOEM-1 N-His |
| Commercial assay or kit | Silica Beads (10 μm) | Corpuscular | C-SIO-10.0 | 10 μm standard microspheres for microscopy |
| Commercial assay or kit | Ni-NTA Agarose | Qiagen | | |
| Commercial assay or kit | Glutathione Sepharose 4B Resion | GE | | |
| Commercial assay or kit | BCA assay | Thermo Scientific | 23225 | |
| Other | GTP | Sigma | 10106399001 | |
| Other | Cholesterol (ovine wool) | Avanti | 700000 | |
| Other | 18:1 (Δ9-Cis) PC (DOPC) (1,2-dioleoyl-sn-glycero-3-phosphocholine) | Avanti | 850375 | |
| Other | C18(Plasm)—18:1 PC (1-(1Z-octadecenyl)—2-oleoyl-sn-glycero-3-phosphocholine) | Avanti | 852467 | |
| Other | Sphingomyelin (Egg, Chicken) | Avanti | 860061 | |
| Other | GM3 Ganglioside (Milk, Bovine-Ammonium Salt) | Avanti | 860058 | |
| Other | 18:1 PS (DOPS) (1,2-dioleoyl-sn-glycero-3-phospho-L-serine (sodium salt)) | Avanti | 840035 | |
| Other | 18:1 (Δ9-Cis) PE (DOPE) (1,2-dioleoyl-sn-glycero-3-phosphoethanolamine) | Avanti | 850725 | |
| Other | C18(Plasm)—18:1 PE (1-(1Z-octadecenyl)—2-oleoyl-sn-glycero-3-phosphoethanolamine) | Avanti | 852758 | |
| Other | Phosphatidylinositol 3-phosphate diC16 (PI(3)P diC16) | Echelon | P-3016 | |
| Other | DiD [DilC18(5); 1,1'-dioctadecyl-3,3,3',3'-tetramethylindodicar-bocyanine, 4-chlorobenzenesulfonate salt] | Thermo Fischer | D7757 | |

## Cloning

Rab5, Rabex5, Rabaptin5, and GDI were cloned into pOEM series vectors (Oxford Expression Technologies), modified to contain a Human Rhino Virus (HRV) 3C cleavable tag at either the N-or C-terminus, followed by a protease cleavage site (Not1 at N-terminus, Asc1 at C-terminus) for insect (SF9) cell expression. Cleavable tags consisted of either 6x-Histidine (6xHis), for Rab5 and Rabex5, or Glutathione S-Transferase (GST), for GDI and Rabaptin5. In order to monitor membrane association and organization, fluorescent Rab5 and Rabaptin5 constructs were created. The proteins were

cloned into SF9 expression vectors containing either an N or C-terminal fluorescent tag (GFP or RFP) attached to the protein by a 13 amino acid flexible linker (N-terminal linker: GSAGSAAGSGAAA; C-terminal: linker: GAPGSAGSAAGSG). As the addition of a fluorescent tag to a protein always carries the risk of altering protein behavior by interfering with protein folding, fluorescent proteins were compared to non-fluorescent constructs known to fold properly by Hydrogen Deuterium Exchange Mass Spectrometry (HDX-MS) and discarded if they showed any aberrant dynamics. The following constructs were used in this study: 6xHis-GFP-Rab5, GST-GDI, GST-Rabaptin5, Rabex5-6xHis, RFP-Rabaptin5, 6xHis-RabexCAT.

## Protein expression and purification

SF9 cells were grown in ESF921 media (Expression Systems) and co-transfected with linearised viral genome and expression plasmid. P1 and P2 virus was generated per manufacturers protocol and yield was optimised by expression screens and infection time course experiments. The P2 virus was used to infect SF9 cells (grown to a density of 1 million cells/ml) at 1% (v/v). Rabex5/Rabaptin5 and geranylgeranylatedRab5/GDI complexes were produced by co-infection. Cells were harvested after 30–40 hr by spinning in a tabletop centrifuge at 500 g for 10 min. Cell pellets were resuspended in Standard Buffer (20 mM Tris pH7.5, 150 mM NaCl, 5 mM MgCl2, 0.5 mM TCEP; STD) supplemented with DNAse one and protease inhibitor cocktail (chymostatin 6 µg/mL, leupeptin 0.5 µg/mL, anti-pain-HCl 10 µg/mL, aprotinin 2 µg/mL, pepstatin 0.7 µg/mL, APMSF 10 µg/mL). Pellets were flash frozen and stored at −80◦C. All subsequent steps performed at 4◦C or on ice. Cells were thawed on ice and lysed by sonication (previously frozen SF9 cell pellets were not sonicated as freeze-thawing is sufficient for lysis). Cell lysates were spun with a JA 25.50 rotor at 22500 rpm for 20 min at 4◦C. Histidine-tagged proteins were bound to Ni-NTA Agarose resin (1L of culture = 1 mL resin) in the presence of 20 mM Imidazole. Resin was washed with STD buffer supplemented with 20 mM Imidazole. Proteins were eluted using 200 mM Imidazole only followed by Histidine-tag cleavage during overnight dialysis with 3C protease. GST tagged proteins were bound to Glutathione Sepharose resin (GS-4B, GE Healthcare) for 2 hr at 4◦C, washed with Standard Buffer and cleaved from resin overnight with a GST-3C protease. Rabex5/Rabaptin5 and Rab5/GDI complexes were purified by both His- and GST-tag affinity purification to obtain pure complex. Size Exclusion Chromatography was performed in STD on a Superdex200 Increase 10/30. Concentrations were determined by a bicinchoninic acid protein Assay (Pierce BCA Protein Assay Kit, ThermoFischer) and purity was assessed by SDS-PAGE followed by colloidal Coomassie staining. Proteins were aliquoted, flash frozen in liquid nitrogen and stored at −80◦C.

## Liposome preparation

The lipids listed below were purchased and resuspended in either CHCl3, CHCl3:MeOH (2:1 for GM3) or CHCl3:MeOH:H2O (1:2:0.8 for PI(3)P) as per manufacturer's instructions and stored at −20◦C. To form liposomes, lipids were mixed together and the solvent was evaporated under a stream of nitrogen. Residual solvent was removed by drying under vacuum overnight in a desiccator. Lipids were rehydrated for at 37◦C in SLB Buffer (20 mM TRIS, 150 mM NaCl) and vortexed to form a stock solution of 1 mM lipid. Small unilamellar vesicles (SUV) were prepared by freeze-thaw cycles (10x snap freezing and thawing at 37◦C). Vesicles were stored at −20◦C and sized by sonication before each application. Size distribution of liposome preparations was assessed by Dynamic Light Scattering using a Zetasizer Nano ZSP Malvern.

### EE lipid composition

DOPC(1,2-dioleoyl-sn-glycero-3-phosphocholine):DOPS(1,2-dioleoyl-sn-glycero-3-phospho-L-serine): DOPE(1,2-dioleoyl-sn-glycero-3-phosphoethanolamine):Sphingomyelin:Cholesterol:ethanolamine plasmalogen (1-(1Z-octadecenyl)−2-oleoyl-sn-glycero-3-phosphoethanolamine):choline plasmalogen (1-(1Z-octadecenyl)−2-oleoyl-sn-glycero-3-phosphocholine):GM3: PI(3P) (diC16 Phosphatidylinositol 3-phosphate): DiD [DiIC18(5); 1,1'-dioctadecyl-3,3,3',3'-tetramethylindodicar-bocyanine, 4-chloro-benzenesulfonate salt] (13.8:6.1:6.8:12.6:32.3:12.9:3.6:9:1:0.1) (See *Table 1*).

## PC/PS lipid composition

DOPC(1,2-dioleoyl-sn-glycero-3-phosphocholine): DOPS(1,2-dioleoyl-sn-glycero-3-phospho-L-serine): Phosphatidylinositol 3-phosphate (PI(3P)diC16): DiD [DiIC18(5); 1,1'-dioctadecyl-3,3,3',3'-tetramethylindodicar-bocyanine, 4-chlorobenzenesulfonate salt] (83.95:15:1:0.1) (See *Table 1*).

## MCB preparation

Silica beads (10 μm standard microspheres for microscopy) were coated with a supported lipid bilayer as described (*Neumann et al., 2013*) with minor modifications to ensure a tight lipid membrane. Beads were incubated with either 800 mM NaCl and 250 μM EE liposomes or 375 μM PC/PS liposomes (Z average diameter 100–120 nm by SLS) for 15 min RT on a rotator wheel. MCBs were washed with 1ml H20 and 2 × 1 mL Standard Buffer, centrifuging at 2000rpm for 1 min in a tabletop centrifuge. Membrane integrity was assessed at different time points and after increasing centrifugation steps. MCBs were found to be robust at 13000 rpm washing steps and up to 4 hr at RT. MCBs were consequently used within 3 hr of formation. The formation protocol was adapted for PC/PS membranes in order to produce MCBs with similar amounts of membrane as compared to EE-MCBs in order to make direct comparisons of GFP-Rab5 recruitment.

## Confocal microscopy

Microscopy experiments were performed on either Nikon TiE (manual imaging, for high resolution and 3D reconstructions) or Cell Voyager 7000S (CV7000S) (automated imaging for time lapse experiments). For manual imaging in Nikon TiE, reactions were prepared in an 8-well NuncTM Lab-TekTM Chamber Slide for imaging. Images were acquired with a Nikon TiE equipped with a 100x/1.45NA Plan Apochromat, DIC oil immersion objective, Yokogawa CSU-X1 scan head and Andor DU-897 back-illuminated CCD. Images were acquired with 80 ms exposure at λ 488, 561 and 660 with the following laser intensities: 15% 488; 5% 561; and 2% 660. For automated imaging, reactions were prepared in a Greiner Square bottom 384 well plate. Images were acquired with Cell Voyager 7000S (CV7000S) equipped with a 60x/1.2NA water immersion objective at 30% 488 and 660 laser. Color and illumination corrections were applied though CV7000S software. Imaging support by M. Stöter (TDS, MPI-CBG).

## Image analysis

Intensity quantifications at MCB equators were performed manually in FIJI (*Schindelin et al., 2012*). Beads were segmented manually and intensity values along the surface were extracted by determining line profiles 10 pixels wide along the surface of the bead as defined by DiD signal. Intensity in 488 and 561 was normalized to the intensity in the 660 channel in a pixelwise manner to account for potential differences in membrane amount between beads and lipid compositions. Box and whiskers plots show median (line), 25/75 quartiles (box boundaries), and min/max values (error bars). Unpaired t-tests were performed to test statistical significance.

To quantify size, intensity, and number of domains on MCBs, a novel image analysis pipeline was developed. The pipeline consists of the following steps, illustrated in *Figure 2—figure supplement 1G*:

1. The membrane surface wass extracted by selecting all pixels in the upper 0.5% intensity percentile for the membrane channel. The sphere center and radius were fitted using a linear least-squares solver with the normal residuals as cost function. Next, particles were distributed on the fitted sphere on a latitude-longitude mesh with $3°$ resolution in both azimuthal and polar directions. The particles were thereafter extended from the surface into a narrow band around it by replicating the particles in the radial direction with 1 pixel spacing until a distance of 5 pixels from the surface. Next, GFP-Rab5 intensity values were interpolated from pixels to particles in the narrow band using moment-conserving interpolation schemes (*Monaghan, 1985*). Finally, the particle intensity values were maximum-projected in the radial direction.
2. Uneven background in the tangent space of beads was corrected using a 'rolling ball' algorithm, with a radius of 2 μm (*Sternberg, 1983*).
3. Rab5 domains were segmented using a globally optimal model-based method Squassh (*Rizk et al., 2014*). The segmentation was applied on pixels after replacing all pixel values with intensities obtained by interpolating back from the particles to pixels to yield a clean pixel

image with denoised and background-corrected intensities. All segmentations were performed using the following parameters in Squassh: regularization parameter 0.35 and minimum object intensity 0.3. In addition, sub-pixel segmentation with 4-fold oversampling was enabled.

4. A marching cubes algorithm (*Lorensen and Cline, 1987*) was used to construct a triangulated mesh of the surface of each segmented domain. Next, the triangulated mesh was used to map the segmentation to the particles in the narrow band. Each particle within the triangulated mesh was orthogonally projected to the surface. For domain size estimation, the projected areas of all in-surface particles belonging to each domain were summed.

## Spatial representation of correlation

Correlation maps were created by computing the normalized mean deviation product (*nMDP*) as a measure of correlation between the corresponding pair of particles with intensities according to the formula:

$$nMDP = \frac{(A_i - \bar{A})(B_i - \bar{B})}{(A_{max} - \bar{A})(B_{max} - \bar{B})}$$

$A_i$ and $B_i$ – intensity of the given particle on the bead $A$ or bead $B$
$\bar{A}$ and $\bar{B}$ – average intensity of the bead $A$ or bead $B$
$A_{max}$ and $B_{max}$ – maximum intensity of the bead $A$ or bead $B$

## Hydrogen Deuterium Exchange-Mass Spectrometry (HDX-MS)

HDX-MS was performed essentially as previously described (*He et al., 2015*; *Mayne et al., 2011*; *Walters et al., 2012*). Proteins (1 uM) are diluted 6:4 with 8M urea, 1% trifluoroacetic acid, passed over an immobilized pepsin column (2.1 mm x 30 mm, ThermoFisher Scientific) in 0.1% trifluoroacetic acid at 15 °C. Peptides are captured on a reversed-phase C8 cartridge, desalted and separated by a Zorbax 300 SB-C18 column (Agilent) at 1 °C using a 5–40% acetonitrile gradient containing 0.1% formic acid over 10 min and electrosprayed directly into an Orbitrap mass spectrometer (LTQ-Orbitrap XL, ThermoFisher Scientific) with a T-piece split flow setup (1:400). Data were collected in profile mode with source parameters: spray voltage 3.4kV, capillary voltage 40V, tube lens 170V, capillary temperature 170 °C. MS/MS CID fragment ions were detected in centroid mode with an AGC target value of $10^4$. CID fragmentation was 35% normalized collision energy (NCE) for 30 ms at Q of 0.25. HCD fragmentation NCE was 35 eV. Peptides were identified using Mascot (Matrix Science) and manually verified to remove ambiguous peptides. For measurement of deuterium uptake, 10 uM protein is diluted 1:9 in Rab5 buffer prepared with deuterated solvent. Samples were incubated for varying times at 22 °C followed by the aforementioned digestion, desalting, separation and mass spectrometry steps. The intensity weighted average m/z value of a peptide's isotopic envelope is compared plus and minus deuteration using the HDX workbench software platform. Individual peptides are verified by manual inspection. Data are visualized using Pymol. Deuterium uptake is normalized for back-exchange when necessary by comparing deuterium uptake to a sample incubated in 6M urea in deuterated buffer for 12–18 hr at room temperature and processed as indicated above.

## Acknowledgements

We warmly thank David H Murray for training in lipid techniques, discussions and support during the initial stages of the project. We also thank Yannis Kalaidzidis, Robert Ernst, Ünal Coskun and Stephan Grill for their helpful discussions and suggestions, as well as Martin Stöter for help with the timelapse imaging. We would also like to thank the following Services and Facilities of the Max Planck Institute of Molecular Cell Biology and Genetics for their support: Light Microscopy Facility (LMF), Technology Development Studio (TDS) and the Protein Expression Purification and Characterization (PEPC) Facility. This work was financially supported by the Max Planck Society (MPG) and the Deutsche Forschungsgemeinschaft (DFG, German Research Foundation) – 1) TRR 83 (grant no. 112927078, TP23 M Zerial) and 2) under Germany's Excellence Strategy – EXC-2068–390729961– Cluster of Excellence Physics of Life of TU Dresden.

## Additional information

### Funding

| Funder | Grant reference number | Author |
|---|---|---|
| Deutsche Forschungsge-meinschaft | TRR 83 112927078 | Marino Zerial |
| Deutsche Forschungsge-meinschaft | TRR 83 TP23 | Marino Zerial |
| Max-Planck-Gesellschaft | | Marino Zerial |

The funders had no role in study design, data collection and interpretation, or the decision to submit the work for publication.

### Author contributions

Alice Cezanne, Conceptualization, Data curation, Formal analysis, Investigation, Methodology, Writing - original draft, Project administration, Writing - review and editing; Janelle Lauer, Conceptualization, Data curation, Supervision, Methodology, Writing - original draft, Project administration, Writing - review and editing; Anastasia Solomatina, Resources, Software, Formal analysis, Visualization, Methodology, Writing - original draft, Writing - review and editing; Ivo F Sbalzarini, Conceptualization, Supervision, Methodology, Writing - review and editing; Marino Zerial, Conceptualization, Supervision, Funding acquisition, Methodology, Project administration, Writing - review and editing

### Author ORCIDs

Alice Cezanne (iD) http://orcid.org/0000-0002-6319-9235
Janelle Lauer (iD) http://orcid.org/0000-0003-1412-6766
Marino Zerial (iD) https://orcid.org/0000-0002-7490-4235

### Decision letter and Author response

Decision letter https://doi.org/10.7554/eLife.54434.sa1
Author response https://doi.org/10.7554/eLife.54434.sa2

## Additional files

### Supplementary files

• Transparent reporting form

### Data availability

All data generated or analysed during this study are included in the manuscript and supporting files. Source data files have been provided for Tables 1, 2 and 3.

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
