## [Decision Letter]

**Acceptance summary:**

Rab GTPases are master regulators of membrane trafficking that template the organization and directionality of the secretory and endocytic pathway. Using a reconstitution approach, the authors describe how a molecular feedback loop is created to regulate Rab function in coordination with specific lipids, effector proteins and guanine nucleotide exchange factors. Understanding feedback loops such as that generated in this system is of broad interest and much remains to be learned about how proteins are organized on intracellular membrane surfaces.

**Decision letter after peer review:**

Thank you for submitting your article "A non-linear system patterns Rab5 GTPase on the membrane" for consideration by *eLife*. Your article has been reviewed by three peer reviewers, including Suzanne Pfeffer as the Senior and Reviewing Editor. The reviewers have discussed the reviews with one another and the Reviewing Editor has drafted this decision to help you prepare a revised submission.

This is a high-quality study in which the authors reconstitute prenylated Rab5 GTPase on bead bound liposomes and reveal macromolecular clustering that depends on a Rab5 GEF and GEF-binding effector. Understanding feedback loops such as that generated in this system is of broad interest and much remains to be learned about how proteins are organized on intracellular membrane surfaces. The surprising result here is that Rab5-GTP alone does not organize but in the presence of the GEF and effector, it does, and the PI3P composition of the membrane is also important.

All the reviewers were highly supportive of the manuscript but suggested various approaches to enhance the significance of the story without too much additional work. We leave it to you to decide how to accomplish this, thus we have included all the reviews--but it would make sense to characterize more fully how PI3P works and add some data more related to lipid composition.

Reviewer #1:

This is a high-quality study in which the authors reconstitute prenylated Rab5 GTPase on bead bound liposomes and reveal macromolecular clustering that depends on a Rab5 GEF and GEF-binding effector. Understanding feedback loops such as that generated in this system is of broad interest and much remains to be learned about how proteins are organized on intracellular membrane surfaces. The surprising result here is that Rab5-GTP alone does not organize but in the presence of the GEF and effector, it does, and the PI3P composition of the membrane is also important. Although there is a large amount of work presented herein, additional experiments would add greatly to the overall impact of the current story. For example, although others proposed in MD simulations a role for an arginine residue in the interaction between Rab5 and PI3P, the authors could easily test this in their system. Not explored is the issue of a catalytic GEF linked to a Rabaptin-5 stoichiometric binding site, suggesting that in this simple system, the GEF may continue to act on a single Rab. This should be discussed more fully and the authors should use standard fluorescent proteins to try to determine the number of molecules in their liposome-bound complexes. What fraction of Rabex is Rabaptin bound in cells, and is it in equilibrium with a soluble pool? If they dimerize the Rabaptin with GFP, do they get twice as large a cluster? And why is there such a great difference between 500nM and 1µM GDI? Is it there to rescue some prenylated proteins that crash out of the system if not able to jump into the membrane? Is there an effect of PRA1 that is so essential for their in vitro endosome fusion reaction? Finally, the authors were not able to remove cholesterol from their system but they could use cholesterol binding protein toxins to segregate cholesterol and look at the effect?

In summary, a slightly deeper consideration of each of the take home messages will add much to this interesting and high-quality study.

Reviewer #2:

In this paper, the authors have used the purified Rabex5/Rabaptin5 complex and supported lipid bilayers to study the association of Rab5 with the bilayer. They report that Rab5 forms domains at the surface of the membrane, that Rabex5 hands Rab5 over to Rabaptin5 upon nucleotide exchange, and that a minimal system consisting of Rab5, RabGDI, and Rabex5/Rabaptin5 is necessary to pattern Rab5 into membrane domains. They also report that early endosomal lipids were required for Rab5 pattern formation. The authors conclude that the prevalence of GEF/effector coupling in nature suggests the existence of a universal system for small GTPase patterning.

The paper is very interesting, because the notion that Rab5, or other small GTPases, forms functional effector domains has broad physiological implications, and was proposed a long time ago by the authors. This is the first, direct in vitro evidence of GTPase domain formation because of a dynamic activation loop controlled by the GEF-GAP-GDI. The study is globally very convincing, but I have a few comments that the authors may wish to address

1) The patterns observed by the authors are dynamically stable. While in a dynamic Turing model of patterning, some area of the phase diagram form stable patterns, it is more likely to obtain travelling waves/spirals in such systems. Did the author observe any propagation or dynamic change of patterns through time? Otherwise, can the author justify using the known time constants or proper feedback loops in their system why it is expected to have a stable pattern rather than travelling waves?

2) The formation of such patterns should strongly depend on the diffusion coefficient of lipids (and proteins), which, I guess is pretty limited onto the supported bilayers used in this study. Did the author try to used GUVs instead of the beads? I would expect that the domains do not form, or that they transform into travelling waves. It would be an important point to make about the role of diffusion in the formation of domains. Diffusion of membrane components on the beads and on the GUVs could also be measured by FRAP, to have an idea of the diffusion change.

3) To reconstitute Rab5 domain formation in vitro, the authors used the lipid composition of enriched early endosomes previously determined by mass spectrometry in Perini (2012). I find it surprising that PC corresponds only to 14 mol% of EE lipids (see Materials and methods). As far as I can remember, PC accounts for 45-55 mol% of all cellular membranes in higher eukaryotes. Also, Perini (2012) is a thesis from the MPI-CBG in Dresden and is not readily available. Sufficient information on fractionation and analysis should be made available in the paper so that readers can evaluate the data. Similarly, the pseudocylindrical projection used to segment and visualized EE-membrane-coated beads is described in a submitted paper (Solomatina et al.).

4) Along the same lines, and due to the high content in sphingolipids and cholesterol in the EE lipid composition used by the authors, how can the authors rule out that Rab5 domains are not formed onto pre-existing lipid domains, in particular because they see domains forming only with the EE composition?

5) I find it a bit difficult to evaluate the projections in Figure 5 A-B, after recruitment of GG-GFP-Rab5 to EE and PC/PS bilayers (without Rabex5/Rabaptin5). I agree with the authors that GFP-Rab5 distribution without Rabex5/Rabaptin5 looks different from the patterns with domains observed in the presence of Rabex5/Rabaptin5 (Figure 6B). Yet, GFP-Rab5 without Rabex5/Rabaptin5 does not seem evenly or randomly distributed on the surface with PI3P (both EE or PC/PS): GFP-Rab5 seems to form numerous, smaller domains. Is this so? Are these domains more or less dynamic by FRAP? Also, GFP-Rab5 fluorescence (Figure 5A-B), but not DiD fluorescence (Figure 5—figure supplement 1), seems to be restricted to the projection above the equator.

Reviewer #3:

The manuscript by Cezanne et al. constitutes a significant breakthrough not only for the narrow field of small GTPases, or even for the cell biology as a whole, but also for the broadest interdisciplinary community interested in biological systems from the systems point of view. Cellular pattern formation and morphogenesis is currently of great interest and is very timely because both the experimental tools and the theoretical methods have reached the needed maturity. The senior author was probably the first to propose, already back in 1997, that the interaction between the GTPase effectors and GEFs can generate positive feedback loops capable of breaking spatial symmetry and generating membrane clusters, the simplest structures to form a prepattern for a great variety of much more complex structures. Zerial and colleagues are again ahead of the competition. Formation of clusters of activated GTPases had been theoretically predicted and computationally modeled for over a decade, but the in vitro reconstitution of this process remained out of reach. Recent tremendous success in reconstituting the MinD system in vitro showed how much can be learned from a reliable, reproducible and controllable in vitro system about cellular morphogenesis. Several groups had been able to reconstitute various Rho GTPases in vitro, but nobody attacked the pattern formation problem. Cezanne et al. do exactly this, they convincingly show that all theoretical predictions were correct, a minimal system consisting of a small GTPase, its GDI and the positive feedback molecule (the effector-GEF complex) is fully competent to break spatial symmetry and generate clusters of activated GTPase.

Therefore, I believe the paper is extremely interesting to a broad readership of *eLife*, very timely and novel and, thus, should be published by *eLife* upon some, mostly minor, improvements.

1) The discovery that only the native endocytic lipid mixture (EE), but not the PC/PS artificial membrane, supports the pattern formation is striking but remains unexplained in the manuscript. It is even more surprising given the result that addition of 1% of PI3P rescues the Rab5 recruitment. Although the authors discuss this result at length, the reader is left feeling somehow unsatisfied… I realize that the authors could not remove lipids from the EE without jeopardizing the integrity of the membrane. What about instead modifying/doping the PC/PS mix? One potential explanation of this result is that some specific lipid/ electric charge is required to make a functional membrane. The authors could "add back" some of the natively found lipids (using their lipidomics analysis of the EE mix as a guide) or increase the negative charge by adding more PS and see if this rescues pattern formation. There is also another potential explanation, which is more interesting from the theoretical point of view and the authors could confirm or reject this possibility with the tools/methods they already have in place! Modeling work predicted and some experiments (Bruurs et al., 2017, 2018, the authors should cite these very important papers!) confirmed that pattern formation is very sensitive to the diffusion coefficient of the GTPase on the membrane. The manuscript shows that the authors used oleic acid derivatives for both PC and PS. While this lipid tail is long, it is also unsaturated, which likely supports liquid disordered lipid phase. To change the diffusion coefficient without affecting the chemistry of lipid head groups, the authors could use longer and fully saturated fatty acid derivatives. Addition of cholesterol to this mix should even further increase the thickness of the membrane bilayer and reduce the diffusion coefficient of Rab5. These experiments are doable without any new protein constructs and could significantly increase the impact of the paper.

2) Another suggestion organically follows from the previous one. The authors site a number of modeling studies but, surprisingly, offer no computational model of their dramatic pattern-forming system themselves! A model could increase the impact of the work and assist in understanding the role of molecular transport (GDI) of Rab5, membrane diffusion coefficient, etc.

---

## [Author Response]

All the reviewers were highly supportive of the manuscript, but suggested various approaches to enhance the significance of the story without too much additional work. We leave it to you to decide how to accomplish this, thus we have included all the reviews--but it would make sense to characterize more fully how PI3P works and add some data more related to lipid composition.Reviewer #1:This is a high-quality study in which the authors reconstitute prenylated Rab5 GTPase on bead bound liposomes and reveal macromolecular clustering that depends on a Rab5 GEF and GEF-binding effector. Understanding feedback loops such as that generated in this system is of broad interest and much remains to be learned about how proteins are organized on intracellular membrane surfaces. The surprising result here is that Rab5-GTP alone does not organize but in the presence of the GEF and effector, it does, and the PI3P composition of the membrane is also important. Although there is a large amount of work presented herein, additional experiments would add greatly to the overall impact of the current story. For example, although others proposed in MD simulations a role for an arginine residue in the interaction between Rab5 and PI3P, the authors could easily test this in their system.

This is an interesting suggestion, unfortunately, due to the Covid-19 crisis which forced us to close the institute, we were unable to complete these experiments for inclusion in this manuscript. We will pursue this as a part of our follow-up work further dissecting the mechanisms controlling Rab5 domain formation.

Not explored is the issue of a catalytic GEF linked to a Rabaptin-5 stoichiometric binding site, suggesting that in this simple system, the GEF may continue to act on a single Rab. This should be discussed more fully and the authors should use standard fluorescent proteins to try to determine the number of molecules in their liposome-bound complexes. What fraction of Rabex is Rabaptin bound in cells, and is it in equilibrium with a soluble pool? If they dimerize the Rabaptin with GFP, do they get twice as large a cluster?

We improved the description of the stoichiometry of the Rabex5/Rabaptin5 complex to make it clearer that it is a heterotetramer composed of two Rabex5 and two Rabaptin5 molecules (Introduction). Rabex5 and Rabaptin5 form a constitutive complex that is stable to gel filtration and we do not observe exchange between a soluble pool of subunits. The precise interaction between these molecules has been investigated in a previous study (Lauer et al., 2019). We also amended the Discussion to include an estimate of the number of Rab5 molecules in our reconstituted domains (paragraph four).

And why is there such a great difference between 500nM and 1µM GDI?

In preliminary experiments we found that 1µM GDI was required to fully remove 10nM GFP-Rab5 from EE-MCBs.

Is it there to rescue some prenylated proteins that crash out of the system if not able to jump into the membrane? Is there an effect of PRA1 that is so essential for their in vitro endosome fusion reaction?

Pra1 was previously included by Ohya et al., 2009, to stimulate the delivery of Rab5 to proteoliposomes and render them fusogenic. We did not look at the addition of Pra1 because we found that a minimal system consisting of GDI and the Rabex5/Rabaptin5 complex was sufficient to create Rab5 domains.

Finally, the authors were not able to remove cholesterol from their system but they could use cholesterol binding protein toxins to segregate cholesterol and look at the effect?

Pra1 was previously included by Ohya et al., 2009, to stimulate the delivery of Rab5 to proteoliposomes and render them fusogenic. We did not look at the addition of Pra1 because we focused on creating a minimal system consisting of GDI and the positive feedback loop provided by the Rabex5/Rabaptin5 complex that was sufficient to create Rab5 domains. In such minimal system the requirement for Pra1 can be bypassed.

Reviewer #2:In this paper, authors have used the purified Rabex5/Rabaptin5 complex and supported lipid bilayers to study the association of Rab5 with the bilayer. They report that Rab5 forms domains at the surface of the membrane, that Rabex5 hands Rab5 over to Rabaptin5 upon nucleotide exchange, and that a minimal system consisting of Rab5, RabGDI, and Rabex5/Rabaptin5 is necessary to pattern Rab5 into membrane domains. They also report that early endosomal lipids were required for Rab5 pattern formation. The authors conclude that the prevalence of GEF/effector coupling in nature suggests the existence of a universal system for small GTPase patterning.The paper is very interesting, because the notion that Rab5, or other small GTPases, forms functional effector domains has broad physiological implications, and was proposed a long time ago by the authors. This is the first, direct in vitro evidence of GTPase domain formation because of a dynamic activation loop controlled by the GEF-GAP-GDI. The study is globally very convincing, but I have a few comments that the authors may wish to address1) The patterns observed by the authors are dynamically stable. While in a dynamic Turing model of patterning, some area of the phase diagram form stable patterns, it is more likely to obtain travelling waves/spirals in such systems. Did the author observe any propagation or dynamic change of patterns through time? Otherwise, can the author justify using the known time constants or proper feedback loops in their system why it is expected to have a stable pattern rather than travelling waves?

We investigated this possibility but did not observe wave-like or other macroscopic dynamic behaviors in our system. However, we cannot exclude the possibility that our system lacks sufficient resolution to detect waves. We amended the Discussion to include a brief discussion of Turing type pattern formation and how stable patterns rather than travelling waves might be formed/observed in our set-up (paragraph five). We are in the process of working on an in-depth computational model to investigate the mechanisms of pattern formation in this system and possible dynamic behaviors.

2) The formation of such patterns should strongly depend on the diffusion coefficient of lipids (and proteins), which, I guess is pretty limited onto the supported bilayers used in this study. Did the author try to used GUVs instead of the beads? I would expect that the domains do not form, or that they transform into travelling waves. It would be an important point to make about the role of diffusion in the formation of domains. Diffusion of membrane components on the beads and on the GUVs could also be measured by FRAP, to have an idea of the diffusion change.

We have not yet tried to use free-standing membranes as this would necessitate another optimization of the system. However, this is something we would like to investigate in future. A consideration here is that the incorporation of 10 lipid components into GUVs is technically challenging. We chose another approach to achieve the same goal. We instead increased the membrane rigidity of the simple PC/PS lipid composition by adding cholesterol, sphingomyelin and ethanolamine plasmalogen (PlasmPE)(Figure 6F and Table 5). Interestingly, we observed that it is not a global change in membrane diffusivity that allows for pattern formation, but most likely local changes induced by increased acyl chain packing and accommodation of the geranylgeranyl anchor.

3) To reconstitute Rab5 domain formation in vitro, the authors used the lipid composition of enriched early endosomes previously determined by mass spectrometry in Perini (2012). I find it surprising that PC corresponds only to 14 mol% of EE lipids (see Materials and methods). As far as I can remember, PC accounts for 45-55 mol% of all cellular membranes in higher eukaryotes. Also, Perini (2012) is a thesis from the MPI-CBG in Dresden and is not readily available. Sufficient information on fractionation and analysis should be made available in the paper so that readers can evaluate the data. Similarly, the pseudocylindrical projection used to segment and visualized EE-membrane-coated beads is described in a submitted paper (Solomatina et al.).

We added a description of our fractionation protocol and our image analysis pipeline (See Materials and methods) to make this information accessible. The image analysis pipeline is further detailed in a separate submitted publication that we hope will be available soon.

While the total phospholipid content of cells is indeed estimated to be composed of 45-55mol% PC, there are significant variations in the PC content between different internal membrane structures. In our EE lipid composition PC and PlasmPC combined make up around 20% of the total phospholipid. This is in agreement with studies reporting the lipid composition of the plasma membrane which is known to be very similar to that of the early endosome and we now comment on this in the Results (subsection “Reconstituting Rab5 domain formation in vitro”). Other endo-membranes, e.g. later steps in the endocytic pathway or the ER, on the other hand have been shown to contain a higher percentage of PC (Casares et al., 2019).

4) Along the same lines, and due to the high content in sphingolipids and cholesterol in the EE lipid composition used by the authors, how can the authors rule out that Rab5 domains are not formed onto pre-existing lipid domains, in particular because they see domains forming only with the EE composition?

This is a plausible interpretation, especially in light of our new findings that cholesterol and sphingomyelin appear to be necessary for Rab5 domain formation. In order to test this, we stained EE-MCBs with C-laurdan. However, we could not detect pre-existing phases of lipid order.

5) I find it a bit difficult to evaluate the projections in Figure 5 A-B, after recruitment of GG-GFP-Rab5 to EE and PC/PS bilayers (without Rabex5/Rabaptin5). I agree with the authors that GFP-Rab5 distribution without Rabex5/Rabaptin5 looks different from the patterns with domains observed in the presence of Rabex5/Rabaptin5 (Figure 6B). Yet, GFP-Rab5 without Rabex5/Rabaptin5 does not seem evenly or randomly distributed on the surface with PI3P (both EE or PC/PS): GFP-Rab5 seems to form numerous, smaller domains. Is this so? Are these domains more or less dynamic by FRAP? Also, GFP-Rab5 fluorescence (Figure 5A-B), but not DiD fluorescence (Figure 5—figure supplement 1), seems to be restricted to the projection above the equator.

We agree that the images originally chosen highlighted inhomogeneity in GFP-Rab5 distribution that might be misleading. To remove a reliance on visual inspection to detect domains, we use automated image analysis methods to detect the presence of domains. We substituted the images in question with other images from the same condition, which are in better agreement with the results from the automated detection algorithm.

Reviewer #3:The manuscript by Cezanne et al. constitutes a significant breakthrough not only for the narrow field of small GTPases, or even for the cell biology as a whole, but also for the broadest interdisciplinary community interested in biological systems from the systems point of view. Cellular pattern formation and morphogenesis is currently of great interest and is very timely because both the experimental tools and the theoretical methods have reached the needed maturity. The senior author was probably the first to propose, already back in 1997, that the interaction between the GTPase effectors and GEFs can generate positive feedback loops capable of breaking spatial symmetry and generating membrane clusters, the simplest structures to form a prepattern for a great variety of much more complex structures. Zerial and colleagues are again ahead of the competition. Formation of clusters of activated GTPases had been theoretically predicted and computationally modeled for over a decade, but the in vitro reconstitution of this process remained out of reach. Recent tremendous success in reconstituting the MinD system in vitro showed how much can be learned from a reliable, reproducible and controllable in vitro system about cellular morphogenesis. Several groups had been able to reconstitute various Rho GTPases in vitro, but nobody attacked the pattern formation problem. Cezanne et al. do exactly this, they convincingly show that all theoretical predictions were correct, a minimal system consisting of a small GTPase, its GDI and the positive feedback molecule (the effector-GEF complex) is fully competent to break spatial symmetry and generate clusters of activated GTPase.Therefore, I believe the paper is extremely interesting to a broad readership of eLife, very timely and novel and, thus, should be published by eLife upon some, mostly minor, improvements.1) The discovery that only the native endocytic lipid mixture (EE), but not the PC/PS artificial membrane, supports the pattern formation is striking but remains unexplained in the manuscript. It is even more surprising given the result that addition of 1% of PI3P rescues the Rab5 recruitment. Although the authors discuss this result at length, the reader is left feeling somehow unsatisfied… I realize that the authors could not remove lipids from the EE without jeopardizing the integrity of the membrane. What about instead modifying/doping the PC/PS mix? One potential explanation of this result is that some specific lipid/ electric charge is required to make a functional membrane. The authors could "add back" some of the natively found lipids (using their lipidomics analysis of the EE mix as a guide) or increase the negative charge by adding more PS and see if this rescues pattern formation.

As the simple PC/PS lipid composition already includes 15mol% DOPS which is a significant amount of negative charge, we modified the PC/PS lipid composition by sequentially adding PI(3)P, cholesterol and either ethanolamine plasmalogen (PlasmPE) or sphingomyelin (the next most abundant lipids in the EE lipid composition which both influence membrane rigidity). We show that the presence of sphingomyelin, but not PlasmPE, is necessary for Rab5 domain formation (Figure 6F and Table 5). We conclude that lipids that are known to mediate acyl chain packing (cholesterol, sphingomyelin) are vital for formation and stability of the Rab5 domain. These points have been expanded in the discussion as well.

There is also another potential explanation, which is more interesting from the theoretical point of view and the authors could confirm or reject this possibility with the tools/methods they already have in place! Modeling work predicted and some experiments (Bruurs et al., 2017, 2018, the authors should cite these very important papers!) confirmed that pattern formation is very sensitive to the diffusion coefficient of the GTPase on the membrane.

The papers mentioned are indeed very interesting and we now cite these papers in the Discussion.

The manuscript shows that the authors used oleic acid derivatives for both PC and PS. While this lipid tail is long, it is also unsaturated, which likely supports liquid disordered lipid phase. To change the diffusion coefficient without affecting the chemistry of lipid head groups, the authors could use longer and fully saturated fatty acid derivatives. Addition of cholesterol to this mix should even further increase the thickness of the membrane bilayer and reduce the diffusion coefficient of Rab5. These experiments are doable without any new protein constructs and could significantly increase the impact of the paper.

We tried to, either fully or partially, replace DOPC(18:1) and DOPS(18:1) with the saturated DSPC(18:0) and DSPS(18:0) species, either fully or partially. Unfortunately, the addition of saturated DSPC and DSPS prevented the formation of stable MCBs in our set up as we conduct experiments at 25°C, well below the T_m_ of DSPC (55°C) and DSPS (68°C). Instead, we addressed the question of Rab5 diffusion by modifying the simple DOPC/DOPS lipid composition to include cholesterol and other components of the EE lipid composition that influence membrane rigidity, as described above.

In the future, we aim to explore the effect of altering lateral diffusion of Rab5 in the membrane in the framework of the computational model described below.

2) Another suggestion organically follows from the previous one. The authors site a number of modeling studies but, surprisingly, offer no computational model of their dramatic pattern-forming system themselves! A model could increase the impact of the work and assist in understanding the role of molecular transport (GDI) of Rab5, membrane diffusion coefficient, etc.

We have been working on an in-depth computational model of this system. This will be submitted as a separate manuscript as it is a full study on its own. The model will delve into the roles of protein-protein, protein-lipid and lipid-lipid interactions highlighted in this study.